# Seasonal, interannual and decadal variability of tropospheric ozone in the North Atlantic: comparison of UM-UKCA and remote sensing observations for 2005–2018

**Maria Rosa Russo**[1,2], **Brian John Kerridge**[3,4], **Nathan Luke Abraham**[1,2], **James Keeble**[1,2], **Barry Graham Latter**[3,4], **Richard Siddans**[3,4], **James Weber**[5,a], **Paul Thomas Griffiths**[1,2], **John Adrian Pyle**[1,2], **and Alexander Thomas Archibald**[1,2]

[1]NCAS National Centre for Atmospheric Science, Cambridge, UK
[2]Department of Chemistry, University of Cambridge, Cambridge, UK
[3]Remote Sensing Group, STFC Rutherford Appleton Laboratory, Didcot, UK
[4]NERC National Centre for Earth Observation, Rutherford Appleton Laboratory, Didcot, UK
[5]Department of Biosciences, University of Sheffield, Sheffield, UK
[a]formerly at: Department of Chemistry, University of Cambridge, Cambridge, UK

**Correspondence:** Maria Rosa Russo (mrr32@cam.ac.uk)

**Abstract.** Tropospheric ozone is an important component of the Earth system as it can affect both climate and air quality. In this work, we use observed tropospheric column ozone derived from the Ozone Monitoring Instrument (OMI) and Microwave Limb Sounder (MLS) OMI-MLS, in addition to OMI ozone retrieved in discrete vertical layers, and compare it to tropospheric ozone from UM-UKCA simulations (which utilize the Unified Model, UM, coupled to UK Chemistry and Aerosol, UKCA). Our aim is to investigate recent changes (2005–2018) in tropospheric ozone in the North Atlantic region, specifically its seasonal, interannual and decadal variability, and to understand what factors are driving such changes. The model exhibits a large positive bias (greater than 5 DU or $\sim 50\%$) in the tropical upper troposphere: through sensitivity experiments, time series correlation, and comparison with the Lightning Imaging Sensor and Optical Transient Detector lightning flash dataset, the model positive bias in the tropics is attributed to shortcomings in the convection and lightning parameterizations, which overestimate lightning flashes in the tropics relative to mid-latitudes. Use of OMI data, for which vertical averaging kernels and a priori information are available, suggests that the model negative bias (6–10 DU or $\sim 20\%$) at mid-latitudes, relative to OMI-MLS tropospheric column, could be the result of vertical sampling. Ozone in the North Atlantic peaks in spring and early summer, with generally good agreement between the modelled and observed seasonal cycle. Recent trends in tropospheric ozone were investigated: whilst both observational datasets indicate positive trends of $\sim 5\%$ and $\sim 10\%$ in North Atlantic ozone, the modelled ozone trends are much closer to zero and have large uncertainties. North Atlantic ozone interannual variability (IAV) in the model was found to be correlated to the IAV of ozone transported to the North Atlantic from the stratosphere ($R = 0.77$) and emission of $NO_x$ from lightning in the tropics ($R = 0.72$). The discrepancy between modelled and observed trends for 2005–2018 could be linked to the model underestimating lower stratospheric ozone trends and associated stratosphere to troposphere transport. Modelled tropospheric ozone IAV is driven by IAV of tropical emissions of $NO_x$ from lightning and IAV of ozone transport from the stratosphere; however, the modelled and observed IAV differ. To understand the IAV discrepancy we investigated how modelled ozone and its drivers respond to large-scale modes of variability. Using OMI height-resolved data and model idealized tracers, we were able to identify stratospheric transport of ozone into the troposphere as the main driver of the

dynamical response of North Atlantic ozone to the Arctic Oscillation (AO) and the North Atlantic Oscillation (NAO). Finally, we found that the modelled ozone IAV is too strongly correlated to the El Niño–Southern Oscillation (ENSO) compared to observed ozone IAV. This is again linked to shortcomings in the lightning flashes parameterization, which underestimates (overestimates) lightning flash production in the tropics during positive (negative) ENSO events.

## 1  Introduction

Ozone ($O_3$) is an important reactive gas present in both the troposphere and the stratosphere. In the stratosphere, ozone is mainly produced following the photolysis of oxygen molecules ($O_2$) by solar ultraviolet radiation. Tropospheric ozone is a greenhouse gas and an oxidant; it can therefore affect climate directly, through its radiative impact, and indirectly, through the oxidation of aerosol precursors and changes to the aerosols' radiative impact (Karset et al., 2018). Tropospheric ozone is formed by photochemical oxidation of volatile organic compounds (VOCs) in the presence of nitrogen oxides ($NO_x = NO + NO_2$) and sunlight; VOCs and $NO_x$ are known as "ozone precursors" (Archibald et al., 2020a; Monks et al., 2015). The increase in anthropogenic emissions of ozone precursors over the last 150 years has led to an estimated 40 % increase in the burden of tropospheric ozone (Archibald et al., 2020a; Griffiths et al., 2021; Young et al., 2013) as simulated by chemistry–climate models (CCMs). As well as being chemically produced in the troposphere, a large fraction of tropospheric ozone comes from downward transport of ozone-rich stratospheric air masses into the troposphere (Škerlak et al., 2014; Young et al., 2018; Archibald et al., 2020a). Stratosphere to troposphere transport (STT) is particularly important at mid-latitudes, where the descending branches of the Hadley and Ferrel cells cause a general downward motion of lower stratospheric air into the troposphere; this results in local stratospheric ozone transport that peaks in late spring and early summer (Škerlak et al., 2014; Yang et al., 2016; Williams et al., 2019). The extent of the stratospheric contribution to tropospheric ozone has been extensively investigated in recent decades (e.g. Lamarque et al., 1999; Neu et al., 2014; Škerlak et al., 2014; Williams et al., 2019; Abalos et al., 2020); however, recent improvements in diagnostic and modelling tools provide evidence that stratospheric ozone has a significant influence on tropospheric ozone trends (e.g. Griffiths et al., 2020) and interannual variability (Terao et al., 2008; Neu et al., 2014; Liu et al., 2020), with STT estimated to contribute up to $\sim 50\,\%$ of tropospheric ozone in the wintertime extratropics (Williams et al., 2019; Abalos et al., 2020) and projected to play an increasingly important role due to the predicted strengthening of the Brewer–Dobson circulation (Butchart, 2014) and possible future reduction in anthropogenic ozone precursor emissions (Archibald et al., 2020a).

CCM simulations show that the greatest increases in tropospheric ozone since the pre-industrial period occur in the Northern Hemisphere and can be attributed to the dramatic increase in precursor emissions in this region (Young et al., 2013; Griffiths et al., 2021). Observational records are limited and do not allow a complete assessment of the pre-industrial to present-day trends in tropospheric ozone (Tarasick et al., 2019), but isotopic evidence supports the general conclusions from CCM simulations of a significant increase in the tropospheric burden since the pre-industrial period (Yeung et al., 2019). The explosion of in situ and satellite observations since the 1990s has enabled a much more precise understanding of the seasonal and interannual variations of tropospheric ozone. Ozonesondes and aircraft measurements give insight into the vertical structure of ozone, whilst measurements from surface stations provide accurate data on the local scale and have been used to estimate ozone trends (e.g. Cooper et al., 2020). However, in situ measurements are geographically sparse and often sample small scale, local features that can be hard for global climate models to reproduce and attribute. As observational records of ozone from satellite platforms increase in length (now spanning decades), they have become an invaluable tool in trying to understand the global tropospheric ozone budget and trends (Archibald et al., 2020b; Heue et al., 2016; Gaudel et al., 2018; Ziemke et al., 2019). In order to investigate recent tropospheric ozone variability this work uses a combination of different satellite ozone measurements and modelled ozone fields.

The tropospheric column ozone (TCO) derived from the Ozone Monitoring Instrument (OMI) and Microwave Limb Sounder (MLS) is a well-documented (Ziemke et al., 2006) and well-established TCO dataset. The ozone column between the surface and the tropopause is derived by subtraction of the MLS stratospheric column from the OMI total column. This dataset has been extensively used in recent studies as a standard for model ozone evaluation (Martin et al., 2007; Young et al., 2013; Gaudel et al., 2018; Archibald et al., 2020b; Griffiths et al., 2021). One problem with using TCO for model evaluation is that ozone has a large gradient around the tropopause, with very high concentrations in the lower stratosphere. Because of this, small differences between the model and OMI-MLS definitions of the tropopause can lead to significant differences between modelled and observed TCO (Griffiths et al., 2021). Furthermore, because of the way OMI-MLS TCO is derived, it is not possible to cor-

rect for its vertical sensitivity through application of averaging kernels (AKs) to the model data. Although neglect of vertical sensitivity makes model comparison with this dataset quicker and less data intensive (and therefore favoured by modellers in the literature), it could influence the comparison between models and observations.

To address the problems described above, OMI-MLS TCO measurements were complemented with ozone data retrieved from the OMI instrument on discrete vertical layers by the Rutherford Appleton Laboratory (RAL) scheme (based on Miles et al., 2015). Because OMI subcolumns are defined independently of the tropopause, they are not subject to the uncertainty associated with tropopause definition and the lower troposphere is well resolved. Furthermore, differences in vertical sampling between model and OMI subcolumns can be reduced through a fairly simple approach whereby OMI monthly mean, gridded AKs and a priori information are applied to the monthly mean modelled ozone data, similar to Williams et al. (2019). The errors arising from using monthly mean satellite operators on monthly mean modelled data have been investigated by Aghedo et al. (2011), who compared model data to Tropospheric Emissions Spectrometer, TES, data; they found only a small difference ($\sim 1\%$–$2\%$) in zonal mean ozone concentrations using monthly mean data compared to a more complex approach where satellite operators are applied to modelled ozone using 3-hourly data. The OMI lower tropospheric column ozone (LTCO), defined between the surface and 450 hPa (or $\sim 0$–$6$ km), provides a measure of ozone in the lower free troposphere. We also use OMI data retrieved in the following two layers: 450–170 hPa ($\sim 6$–$13$ km) and 170–50 hPa ($\sim 13$–$20$ km); these span the upper troposphere and lower stratosphere and can help to evaluate ozone in the North Atlantic region. Since ozone is not vertically homogeneous in the troposphere, insight can be gained by differentiating between the lower and upper troposphere, where ozone has different sources, sinks and lifetimes (Lelieveld and Dentener, 2000).

The North Atlantic is an interesting region where decadal changes in climate, spanning the atmosphere, ocean and cryosphere, interact to produce periods of faster warming and cooling, known as Atlantic multidecadal variability (AMV, Sutton et al., 2018). The AMV has been linked to a number of local and non-local impacts, such as rainfall anomalies, changes in the frequency of hurricanes and Greenland ice sheet melt, to name just a few (Robson et al., 2018, and references therein). The leading mode of atmospheric variability in the North Atlantic climate system is the North Atlantic Oscillation, which drives interannual variability in tropospheric ozone, temperature and precipitation over Europe (Robson et al., 2018 and references therein). Understanding decadal changes in ozone and its drivers can help us predict future changes in North Atlantic ozone and how to mitigate its impact on, for example, exacerbating air quality problems. The link between the Arctic Oscillation (AO) or North Atlantic Oscillation (NAO) and ozone in the North Atlantic has

long been investigated (Creilson et al., 2003; Lamarque and Hess, 2004; Creilson et al., 2005; Hess and Lamarque, 2007; Pausata et al., 2012; Pope et al., 2018). Surface ozone anomalies over northern Europe were shown to correlate strongly to the NAO (Pausata et al., 2012) and this was attributed to increased westerly flow across the North Atlantic leading to increased transport of pollutants from the US to Europe during a positive NAO phase. However, the response of North Atlantic ozone to the AO and NAO has been shown to vary with height and location. Lamarque and Hess (2004) found a strong correlation across the vertical column between the spring AO and ozonesonde data for the US, while data for Europe showed the strongest correlation with the AO at higher altitudes (with maximum at $\sim 200$ hPa). Similarly, Pope et al. (2018) showed the difference in observed ozone between positive and negative winter NAO phases varies with height and location across the North Atlantic. Pausata et al. (2012) analysed modelled ozone anomalies in the North Atlantic and suggested that whilst surface ozone correlation to the NAO can be explained by long-range transport of ozone and ozone precursors, the downward transport of stratospheric air might play a larger role in the tropospheric column, particularly in winter and spring.

This work is part of a coordinated effort to characterize the climate and composition of the North Atlantic region (Sutton et al., 2018). Recent changes in the North Atlantic climate system have occurred for a number of physical and chemical variables and have been highlighted in Robson et al. (2018). Significant decadal variability has been observed for the North Atlantic Oscillation and the speed of the jet stream (Hurrell, 1995; Woollings et al., 2015), ocean heat and salinity content (Robson et al., 2016; Reverdin, 2010), sea ice extent (Swart et al., 2015), and rate of transport by ocean currents (Smeed et al., 2018). Ozone trends in the North Atlantic can be influenced by a variety of factors. Whilst there is consensus on the long-term increase in global ozone burden, it is harder to pinpoint its magnitude due to the sparse nature and reliability of early ozone measurements. Using isotopic evidence from polar firn and ice and some model simulations, Yeung et al. (2019) estimated an ozone increase of less than 40 % between 1850 and 2005. Tarasick et al. (2019) found surface ozone increases of 30 %–70 % between historical (1877–1975) and present-day (1975–2015) measurements at rural Northern Hemisphere stations; they also found that free tropospheric ozone has increased by $\sim 50\%$ between the same period for northern Europe and the eastern USA. CMIP6 model integrations are consistent with observations, with the multi-model ensemble mean producing an increase in tropospheric ozone burden of $\sim 109 \pm 25$ Tg ($\sim 40\%$) between 1850–1859 and 2005–2014 (Szopa et al., 2021); this change in ozone has been attributed to an increase in anthropogenic ozone precursor emissions over the same time period (Szopa et al., 2021).

In most recent decades, between the mid-1990s and present day, we see a more marked ozone increase in tropi-

cal regions compared to mid-latitudes (Gulev et al., 2021). At northern mid-latitudes, surface and low-altitude ozone trends are variable, with some positive and some negative trends, but more positive values are observed in tropical regions (Cooper et al., 2020; Gaudel et al., 2020), where changes are between 2 % and 17 % per decade (Gulev et al., 2021). Similarly, ozone in the tropical free troposphere has increased more compared to ozone in the mid-latitude free troposphere, with increases of 2 %–12 % per decade and 2 %–7 % per decade, respectively (Cooper et al., 2020; Gaudel et al., 2020; Gulev et al., 2021; Chang et al., 2022).

Anthropogenic emissions of ozone precursors have been decreasing in North America and Europe since the 1990s as a result of air quality policies; this reduction is potentially contributing to lower tropospheric ozone trends at northern mid-latitudes compared to equatorial regions, where anthropogenic emissions of ozone precursors have continued to increase (Archibald et al., 2020a). Due to the relatively long lifetime of free tropospheric ozone, 20–30 d (Young et al., 2013; Monks et al., 2015), North Atlantic ozone concentrations can also be affected by hemispheric transport of ozone generated by emissions outside of the local region (e.g. Butler et al., 2018; Sorooshian et al., 2020). Other potential factors contributing to North Atlantic ozone trends include changes in tropical biogenic and biomass burning emissions, tropical $NO_x$ emissions from lightning, and transport of ozone-rich air from the stratosphere.

Several studies have focused on ozone trends in Europe, the USA, and the North Atlantic region using surface measurements, sondes, aircraft and satellite observations (Cooper et al., 2014; Parrish et al., 2014; Oetjen et al., 2016; Heue et al., 2016; Gaudel et al., 2020; Cohen et al., 2018; Cooper et al., 2020; Chang et al., 2022). However, due to ozone's large interannual variability, calculated trends can be influenced by the reference years; furthermore, due to ozone spatial heterogeneity and large seasonal variations, reported trends can differ in magnitude depending on the horizontal or vertical location and season (e.g. Cohen et al., 2018).

Our focus in this study is to investigate recent changes (2005–2018) in tropospheric ozone in the North Atlantic using satellite observations and a state-of-the-art chemistry climate model, the UK Chemistry and Aerosol (UKCA); UKCA (Archibald et al., 2020b) is the chemistry and aerosol component of the UK Earth System Model, UKESM1 (Sellar et al., 2019). Our aim is to investigate tropospheric ozone seasonal, interannual and decadal variability and to understand what factors are driving such changes. We also aim to investigate the role of large-scale modes of variability, such as the Arctic Oscillation (AO), North Atlantic Oscillation (NAO) and El Niño–Southern Oscillation (ENSO), on North Atlantic ozone interannual variability (IAV). Satellite observations on discrete atmospheric layers are used as an additional benchmark for model evaluation and to improve our understanding of the chemical and dynamical processes affecting tropospheric ozone.

Model configuration, observational datasets and numerical methods used in this paper are described in Sect. 2; in Sect. 3 we analyse observed and modelled tropospheric ozone climatology in the North Atlantic, including seasonal variations, and address possible reasons for the discrepancy between model and observations; in Sect. 4 we discuss tropospheric ozone interannual and decadal variability in the North Atlantic and what drives ozone variability and trends; the conclusions of this work are presented in Sect. 5.

## 2 Technical details

### 2.1 Model description

Model simulations of the UKCA chemistry climate model were performed with a horizontal grid of $1.875° \times 1.25°$ and 85 vertical levels with a model top at 80 km TS1. The specific configuration is a combination of the StratTrop chemistry scheme coupled to Global Atmosphere 7.1 (Walters et al., 2019) and has been described in Archibald et al. (2020b). The UKCA StratTrop scheme merges the stratospheric scheme described in Morgenstern et al. (2009) with the tropospheric "TropIsop" scheme described in O'Connor et al. (2014). UKCA-StratTrop describes the chemical processing of the organic compounds – methane, ethane, propane, isoprene and their oxidation products – coupled to the inorganic chemistry of $O_x$, $NO_x$, $HO_x$, $ClO_x$ and $BrO_x$, including heterogeneous processes on polar stratospheric clouds and liquid sulfate aerosols. For more details on this model and a general model evaluation, the reader is referred to Archibald et al. (2020b) and references therein.

In this work we use nudged model integrations wherein the model meteorology is relaxed toward the ECMWF's ERA-Interim reanalysis (Dee et al., 2011) using the nudging functionality in the MetUM (Telford et al., 2008). Nudging is applied to model temperature and winds from about 1.2 to 65 km (maximum height of ERA data) using an e-folding relaxation timescale of 6 h. CMIP6 emissions (Feng et al., 2020) are used to drive the modelled chemistry; historical emissions are used up to 2014 and SSP3-7.0 from 2015 to 2018.

In order to identify the impact of transport on modelled tropospheric ozone in the North Atlantic, we used two idealized tracers, O3S and O3S-C, which represent ozone transported from the stratosphere and stratosphere to troposphere transport, respectively. The O3S tracer is set to the same values as stratospheric ozone in the stratosphere and decays following ozone chemical loss reactions in the troposphere and has been used in previous studies (e.g. CCMI simulations). The O3S-C is defined similarly to O3S in the troposphere (i.e. decays with the same chemical loss reactions), but its stratospheric concentration is homogeneous in space and constant in time. O3S-C therefore gives a complementary measure of downward transport from the stratosphere that is not affected by stratospheric ozone geographical dis-

tribution or trends. The pair of O3S and O3S-C tracers therefore allow us to quantify the effects of STT on tropospheric ozone and isolate the effects of stratospheric circulation and dynamical trends on STT vs. those from changes in the burden and distribution of lower stratospheric ozone.

## 2.2 Observations

This study uses observations by the Ozone Monitoring Instrument (OMI) and Microwave Limb Sounder (MLS) on NASA's Aura satellite. OMI is a nadir-viewing UV/VIS solar backscatter spectrometer with $13 \times 24$ km horizonal sampling. Spectra in the Huggins bands are used to retrieve total column ozone. The OMI-MLS tropospheric column ozone (TCO) is determined by subtracting the MLS stratospheric column ozone (Waters et al., 2006) from the OMI (Levelt et al., 2006) total column ozone. The algorithm used to produce the tropospheric ozone column is described in Ziemke et al. (2006, 2019), and the monthly gridded data product is available between 60° S and 60° N with a horizontal resolution of $1° \times 1.25°$. The data were downloaded from https://acd-ext.gsfc.nasa.gov/Data_services/cloud_slice/new_data.html in 7 July 2020.

OMI measurements of Hartley and Huggins bands spectra are used to retrieve ozone profiles spanning the stratosphere and troposphere. The height-resolved ozone dataset used in this study was produced by the Remote-Sensing Group at Rutherford Appleton Laboratory (RAL) using a profile retrieval scheme based on a method first developed for the GOME series of instruments (Miles et al., 2015) and applied to produce multi-year datasets from a series of UV/VIS sounders for ESA's Climate Change Initiative and EU's Copernicus Climate Change Service. Surface–450, 450–170 and 170–50 hPa layer amounts from individual soundings were gridded to monthly data with a horizontal resolution of $1.5° \times 1.5°$.

A bias correction, derived with respect to a multi-year ensemble of ozonesondes as a function of latitude and month of year, has been applied to each OMI subcolumn (for more details on the bias correction, see Fig. S1 in the Supplement).

OMI is the first of a new class of UV/VIS sounder which uses 2-D detector arrays rather than scanning 1-D arrays to scan across-track. However, across-track sampling is limited by an obstruction to its field of view (the so-called row anomaly), which changes over the course of the mission and particularly limits sampling in the Northern Hemisphere. The reduction in across-track sampling over time has the largest impact in the northern mid-latitudes and, although it does not seriously affect the multi-year mean ozone distribution, it results in larger uncertainties in the trend estimates for the lower troposphere subcolumn. A measure of this uncertainty is provided in Fig. 6. Other OMI subcolumns are less sensitive to these issues and show much smaller uncertainties in both climatological ozone distribution and trends.

The BSVertOzone (Bodeker Scientific Vertical Ozone) is a global, vertically resolved, monthly mean, zonal mean ozone dataset; this dataset includes data from satellites and ozonesondes and covers the period from 1979 to 2016. For more details the reader is directed to Hassler et al. (2018a). The data were downloaded from https://zenodo.org/record/1217184#.YbchxS-l2X1 in 12 March 2020.

The LIS-OTD dataset combines data from the Optical Transient Detector (OTD) and the Lightning Imaging Sensor (LIS) to measure lightning flash rates on the global scale (Cecil and NASA MSFC, 2006; Cecil et al., 2014) and is provided by the NASA Global Hydrology Resource Center (GHRC). In this study we use version 2.3 of the low-resolution monthly time series (LRMTS, downloaded from https://ghrc.nsstc.nasa.gov/hydro/details/lolrmts in 17 September 2019), which provides monthly gridded data at a resolution of $2.5° \times 2.5°$. OTD flew from 1995 to 2000 on the MicroLab-1 satellite (Christian, 2003). LIS has been deployed on the Tropical Rainfall Measuring Mission (TRMM) satellite from 1997 to 2015 (Bocippio et al., 2002). These space-based optical lightning sensors detect both cloud-to-ground (CG) and cloud-to-cloud (CC) discharges and are well suited for determining how lightning is distributed across the Earth's surface.

## 2.3 Data processing

For comparison with OMI-MLS data, the modelled ozone tropospheric column is calculated by vertically integrating the model ozone between the surface and the tropopause (defined as 380 K + 2 PV). Alternative definitions of tropopause have also been used to address the sensitivity of our results to the choice of tropopause: these include the World Meteorological Organization (WMO) $2 \text{ K km}^{-1}$ thermal vertical gradient and 125 ppbv ozonopause.

To ensure consistent comparison between OMI and UKCA ozone, we used monthly gridded averaging kernels and a priori information to minimize vertical sampling differences between model and observations (similar to Williams et al., 2019). Modelled ozone data were first regridded on the OMI horizontal grid; model grid points for which observational data are not available (due to cloud screening, solar zenith angle and other sampling limitations at high latitudes) were removed; the remaining co-located spatial and temporal grid points (function of latitude, longitude and time) were interpolated vertically to match the pressure levels of the observations, then sampled using the OMI a priori and averaging kernel information (as described in Eq. 1):

$$x^{\text{s}} = x^{\text{a}} + \mathbf{A} \left( x^{\text{m}} - x^{\text{a}} \right), \tag{1}$$

where $x^{\text{s}}$ is the model gridded ozone profile sampled as OMI, $x^{\text{a}}$ is the OMI a priori gridded profile, $x^{\text{m}}$ is the model gridded ozone profile (interpolated on OMI pressure levels) and $\mathbf{A}$ is the gridded OMI averaging kernel matrix. Finally, the

model data were integrated vertically to produce ozone sub-columns consistent with the OMI subcolumns.

Although potential issues with using monthly mean, rather than averaging kernels for individual profiles, can arise for certain species and instruments (von Clarmann and Glatthor, 2019), agreement between model and observations was found to be improved substantially through application of monthly mean averaging kernels in this analysis, in agreement with previous work (Aghedo et al., 2011; Williams et al., 2019).

Trends were calculated using a least-squares linear regression method on monthly deseasonalized time series. The standard error of the trend estimate has been calculated from the standard deviation of the residuals (Wigley et al., 2006). The effect of autocorrelation has been included by using the lag-one autocorrelation coefficient to determine an effective sample size in the calculation of the standard error of the trend estimate (Wigley et al., 2006; Santer et al., 2000).

## 3 Tropospheric ozone climatology: geographical distribution and seasonality

Unless stated otherwise, all plots in this section use data from January 2005 to December 2018. Analysis of TCO from OMI-MLS is supplemented by analysis of LTCO from the lowest OMI subcolumn (surf–450 hPa) and upper tropospheric column ozone (UTCO) from OMI subcolumn 450–170 hPa. Where quantities are presented as a regional average for the North Atlantic, the latitude–longitude coordinates are defined as 0–60° N and 100° W–30° E; for consistency, these are the same coordinates used to plot the regional maps. For line plots showing domain averages we chose to analyse mid-latitude north Atlantic (MNA) between 30 and 60° N and the tropical north Atlantic (TNA) between 0 and 30° N, as well as the North Atlantic domain as a whole; this is because chemical and dynamical drivers of tropospheric ozone in the tropics can be different from those at mid-latitudes.

### 3.1 Observed vs. modelled tropospheric ozone climatology in the North Atlantic

We start our analysis by investigating the geographical distribution of tropospheric ozone in the North Atlantic using multiannual mean maps. Figure 1 shows observed and modelled ozone columns and their absolute and percentage difference for both TCO (left column) and LTCO (right column). Observed TCO (Fig. 1a) shows a local maximum around 30–40° N with generally lower ozone values over the tropical part of the domain; in contrast, UKCA TCO (Fig. 1c) has larger values in the tropical part of the domain, with a pronounced local maximum over Northern Africa, resulting in an overestimate of observed TCO for large parts of the Tropical North Atlantic. Difference maps (Fig. 1e and g) show a significant discrepancy between modelled and observed TCO in the southern North Atlantic, with biases

greater than 10 DU or ∼ 40 % for a large area in the tropics; at mid-latitudes the model underestimates TCO by ∼ 6–10 DU or ∼ 20 %. Comparisons of observed and modelled LTCO are shown in Fig. 1b and d: the model exhibits a relatively small positive bias of ∼ 2–4 DU or ∼ 10 %–20 % (Fig. 1f and h) over a large part of the North Atlantic. This bias is considerably smaller than the model bias in TCO. Williams et al. (2019) compared OMI ozone column in the lower troposphere with EMAC and CMAM models and found a widespread (global) positive bias between EMAC and OMI LTCO.

To understand the seasonality of North Atlantic ozone we analyse multiannual mean seasonal cycles (Fig. 2) averaged over the following domains: North Atlantic (a, b), mid-latitude North Atlantic (MNA) (c, d) and tropical North Atlantic (TNA) (e, f); seasonal cycles for TCO and LTCO are shown in the first and second row, respectively. The amplitude of the seasonal cycle varies in different regions, with larger seasonal changes being observed at mid-latitudes compared to the tropical part of the domain. Observations (black lines) show a broad maximum in spring and early summer for all three regions; this is consistent with previous studies (Logan, 1985; Parrish et al., 2014) and can be attributed to two major sources of tropospheric ozone: transport from the stratosphere (with a maximum in late spring and early summer) and photochemical production from ozone precursors' emissions (with a maximum in summer).

Despite the regional biases highlighted in Fig. 1, the model is generally able to reproduce observed seasonal variations in tropospheric ozone. The seasonal cycle of modelled TCO and LTCO is consistent with observations, especially at mid-latitudes (Fig. 2c, d), where the amplitude and phase of the modelled and observed seasonal cycles are in good agreement. However, UKCA TCO shows an additional seasonal maximum in the tropical North Atlantic in late summer that is not present in the observations (Fig. 2e). This late summer discrepancy is less marked for LTCO (Fig. 2f).

Seasonal maps, shown in Fig. S2 (SF 2), can help identify the regions linked to the seasonal maxima in Fig. 2. In SF 2a, the observed summer maximum in TCO is centred around 30–40° N; in contrast, the modelled summer TCO (SF 2c) exhibits an additional maximum over northern Africa and parts of the tropical Atlantic Ocean. The observed TCO maximum between 30 and 40° N in spring is consistent with stratosphere to troposphere transport (STT) of ozone, which typically occurs around 30° N in the spring and early summer (Škerlak et al., 2014).

From the analysis of Figs. 1, 2 and S2, it appears that modelled LTCO is in generally good agreement with observations, whilst modelled TCO exhibits a large positive bias and an additional seasonal maximum in late summer compared to observations, with both discrepancies occurring over the tropical North Atlantic and northern Africa. These results are not unexpected: recent UKCA model evaluations found a widespread TCO positive bias across most of the tropics

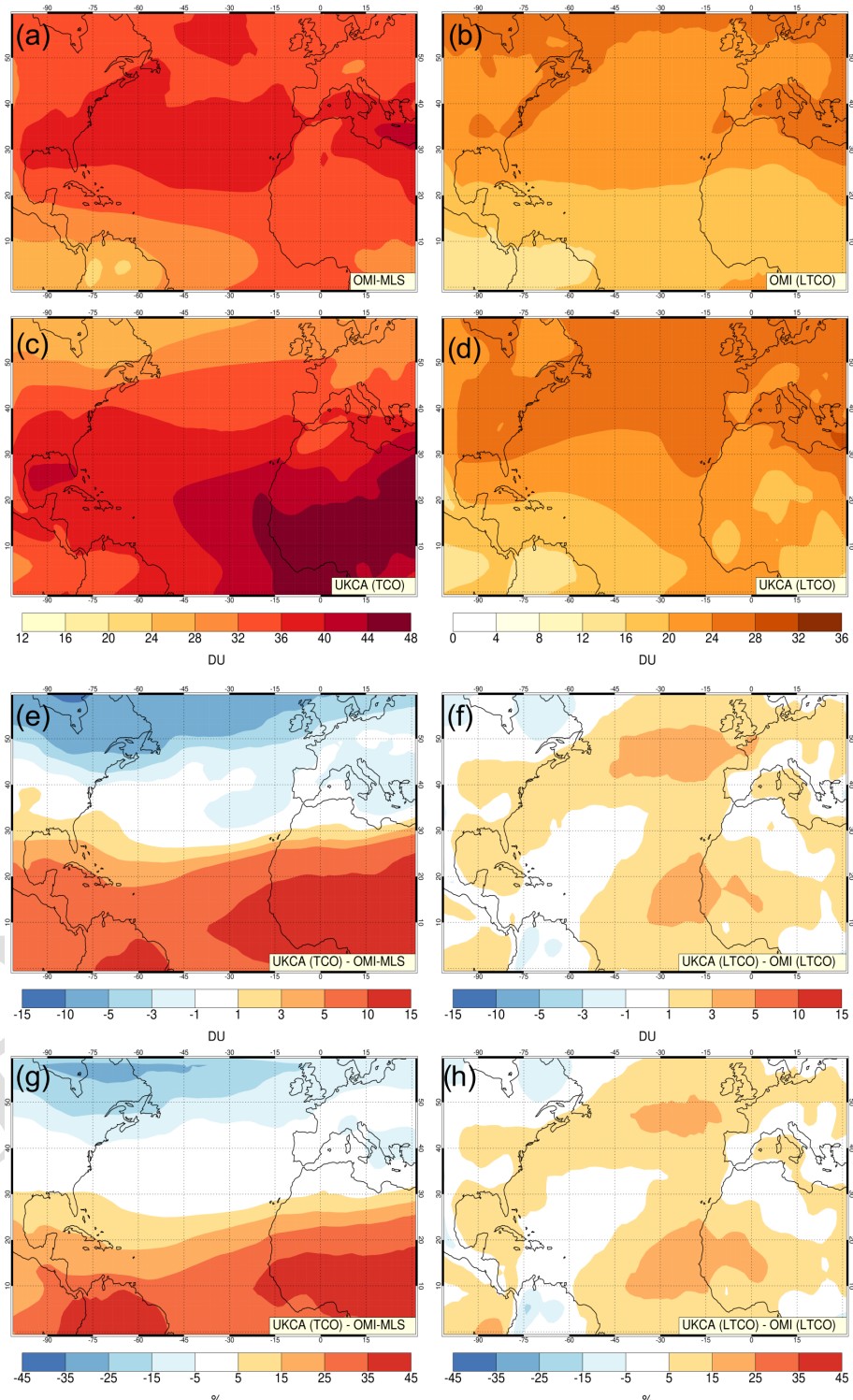

**Figure 1.** Tropospheric ozone multiannual means for the period 2005–2018: **(a)** OMI-MLS TCO, **(b)** OMI LTCO, **(c)** UKCA TCO and **(d)** UKCA LTCO. Difference between modelled and observed results: **(e)** UKCA TCO–OMI-MLS, **(f)** UKCA LTCO–OMI LTCO. Panels **(g)** and **(h)** are the same as panels **(e)** and **(f)** but expressed as a percentage difference.

(Archibald et al., 2020b; Robson et al., 2020) and this bias is mostly unchanged when a more complex chemistry scheme is used (Archer-Nicholls et al., 2021). UKCA is also not unique in exhibiting a TCO bias with respect to OMI-MLS: GFDL AM4.1 (Horowitz et al., 2020) has a widespread positive bias in the tropics which is in excess of 20 % over Africa and South America.

Furthermore, there is also some uncertainty in TCO measurements, which is reflected in the spread of values from different satellite platforms: Gaudel et al. (2018) reported average annual TCO values from five different satellite datasets ranging between ∼ 25 and 35 DU for the latitude band 0–30° N and the period from 2005 to 2016. Despite this spread in the observed TCO values, UKCA TCO, calculated for the same latitude band and period described in Gaudel et al. (2018), shows values in the range 35–39 DU, which are outside the range of uncertainty of the combined observations. Gaudel et al. (2018) reported a mean ozone burden, from five satellite datasets between 60° S and 60° N, of ∼ 300 Tg ± 6 % for the most recent satellite record (up to 2016). In our study the tropospheric ozone burden from OMI-MLS and UKCA for the 2005–2018 period are 297 and 301 Tg respectively. Although UKCA's ozone burden in the 60° S–60° N range shows a good agreement with observations, Archibald et al. (2020b) showed that the UKCA global tropospheric ozone burden is consistent with observations as a result of an overestimate of TCO in the tropics and an underestimate of TCO at mid-latitudes, which is in line with our findings (see also Fig. S4).

One well-known issue when using TCO to evaluate modelled tropospheric ozone is that small differences in tropopause definition can lead to large differences in TCO due to the strong ozone gradient around the tropopause. OMI-MLS data uses the WMO lapse rate tropopause, calculated from NCEP reanalysis, to estimate tropopause pressure and define TCO (Ziemke et al., 2006). The first step is therefore to ascertain whether the modelled positive bias is real and not an artefact of different tropopause definitions being used for the calculation of the tropospheric column in the modelled and observed TCO. Figure S3 (SF 3) shows three modelled TCOs and their respective bias relative to OMI-MLS, calculated for three different tropopause definitions, including the 125 ppb ozonopause and the WMO lapse rate. The modelled TCO varies with different tropopause definitions (similarly to the findings in Griffiths et al., 2021) but the large positive bias in the tropics remains a common feature despite the different tropopause definitions. We can therefore conclude that the modelled positive bias in the tropical North Atlantic is larger than the uncertainty arising from the choice of tropopause used in the calculation of the TCO.

Modelled LTCO shows a smaller bias and a better seasonal agreement with observed OMI data; this could be either because the model bias is smaller in the lower troposphere or because averaging kernels and a priori information from the OMI data were used to construct modelled LTCO,

hence increasing the model's ability to reproduce observations by using the same vertical sampling and cloud screening as the satellite data (Williams et al., 2019). In order to discern between these two possibilities, we compare the modelled and OMI subcolumns between 450 and 170 hPa. This retrieved subcolumn is most sensitive to heights between ∼ 6 and 13 km which, for tropical latitudes, is below the tropopause. Figure 3 shows a comparison of observed and modelled ozone subcolumns for ∼ 450–170 hPa in the tropical North Atlantic; despite the use of satellite AK and a priori information, the model shows a large positive bias in the tropical upper troposphere. The bias is largest over sub-equatorial Africa, and differences are larger than 6 DU (Fig. 3c) or ∼ 60 % (Fig. 3d) for a large part of the tropical domain; this is consistent with the TCO bias shown in Fig. 1e and g. Use of the OMI data for different vertical layers allows us to recognize that the model TCO bias in the tropical North Atlantic results from a large positive bias in the tropical upper troposphere and a smaller bias in the tropical lower troposphere. The tropical bias in UKCA is not restricted to the North Atlantic (Archibald et al., 2020b; Archer-Nicholls et al., 2021). An extension to the global scale is shown in Fig. S4: the top row provides a comparison between zonally averaged ozone vertical profiles from UKCA and the Bodeker Scientific Vertical Ozone dataset (Hassler et al., 2018a); the middle and bottom rows show global multiannual mean maps of differences between modelled and observed ozone in the three separate columns (LTCO, UTCO and TCO), confirming that the model bias extends over a large geographical region of the tropical upper troposphere.

Whilst the modelled tropical bias is supported by the comparison with the BS vertical ozone dataset (SF 4b), with the largest differences in the tropical upper troposphere, the modelled vertical profiles are in generally good agreement with BS observations in the northern and southern mid-latitudes (SF 4a and c), therefore leaving some questions about the modelled TCO negative bias at southern mid-latitudes relative to OMI-MLS (see SF 4f, i). We have shown that when satellite operators are used to correctly sample model data there is generally a good agreement between UKCA and OMI ozone in the lower troposphere (Fig. 1f, h and SF 4d, g); however, when no operators are used to correct for vertical sampling, a negative bias (similar in magnitude to the TCO bias) is detected for the modelled lower tropospheric ozone at mid-latitudes and high latitudes due to stratospheric influence on the observations (not shown). It is possible that the modelled TCO negative bias at southern mid-latitudes and high latitudes and northern high latitudes relative to OMI-MLS might also be the result of inconsistent sampling between model and observations due to the lack of satellite operators for this observational dataset.

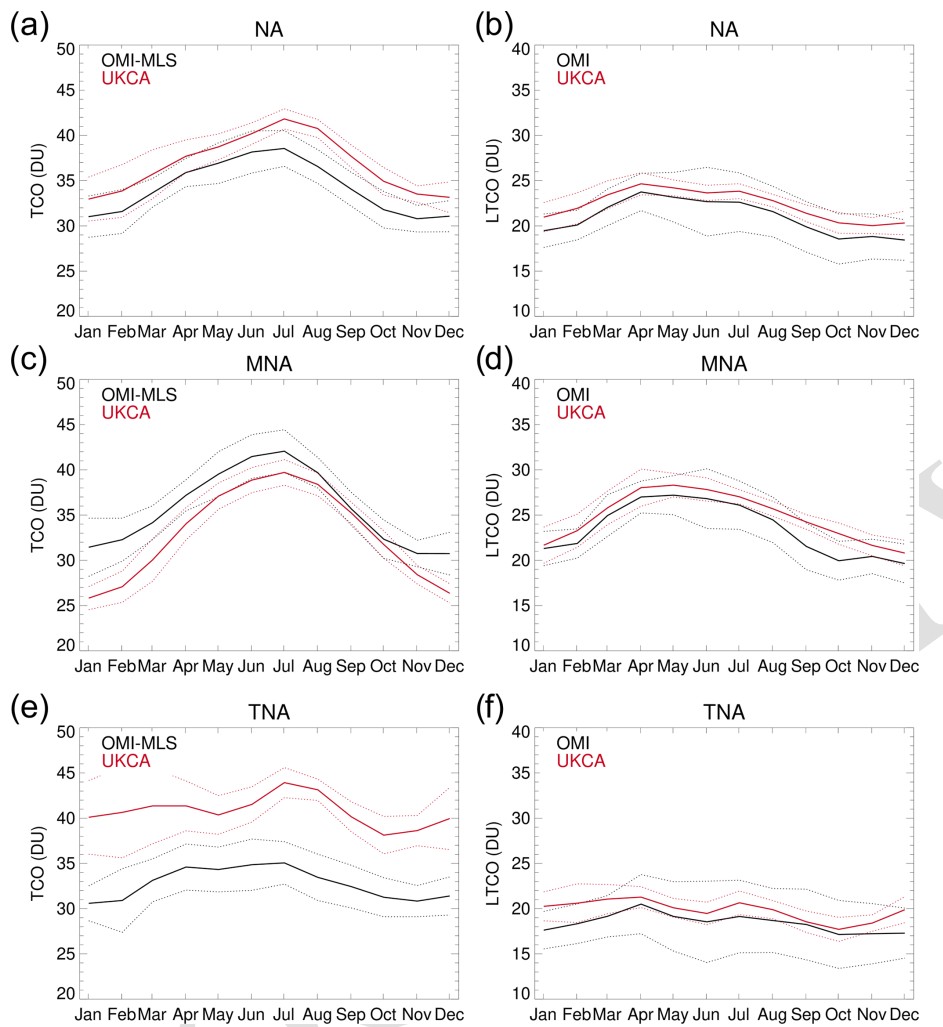

**Figure 2.** Tropospheric ozone seasonal cycle averaged over the North Atlantic **(a, b)**, mid-latitude North Atlantic **(c, d)** and tropical North Atlantic **(e, f)**. The first column shows TCO and second column shows LTCO. Dotted lines indicate the 2 SD (standard deviation) interval.

## 3.2 Sources of uncertainty for modelled tropospheric ozone in the North Atlantic basin

Having established that the model bias is greatest in the tropical upper troposphere, we now focus on possible reasons for such discrepancy. Ozone precursor gases can be emitted from both anthropogenic and natural sources. Present-day anthropogenic emissions are generally well constrained, and their geographical locations, seasonal variations and magnitudes are derived from emission inventories and inverse modelling techniques (Lamarque et al., 2010; Feng et al., 2020). In contrast, some natural emissions of ozone precursors can have quite large uncertainties; these include CO and $NO_x$ from biomass burning, soil $NO_x$, biogenic isoprene, and $NO_x$ from lightning. An overestimate of such ozone precursor emissions in the model could therefore result in an overestimate of tropospheric ozone. Please note that, with the exception of lightning, all other natural and anthropogenic sources of

$NO_x$ at the surface are combined in the model and referred to as surface $NO_x$ emissions.

Biomass burning emissions and biogenic emissions of isoprene are responsible for a large part of VOC emissions in the tropics. The largest source of biomass burning in the North Atlantic is from equatorial Africa (between 0–15° N); these emissions have a marked seasonal maximum in winter (Roberts et al., 2009; van der Werf et al., 2017) and a large interannual variability. However, model sensitivity experiments showed that completely removing biomass burning emissions from the model input has a very small impact on the tropospheric ozone budget, with a maximum differences of ∼ 2 DU in August (Shin, personal communication, 2020); this suggests that whilst biomass burning emissions might be important on the local scale, they do not have a large impact on tropospheric ozone at the regional and global scale in UKCA.

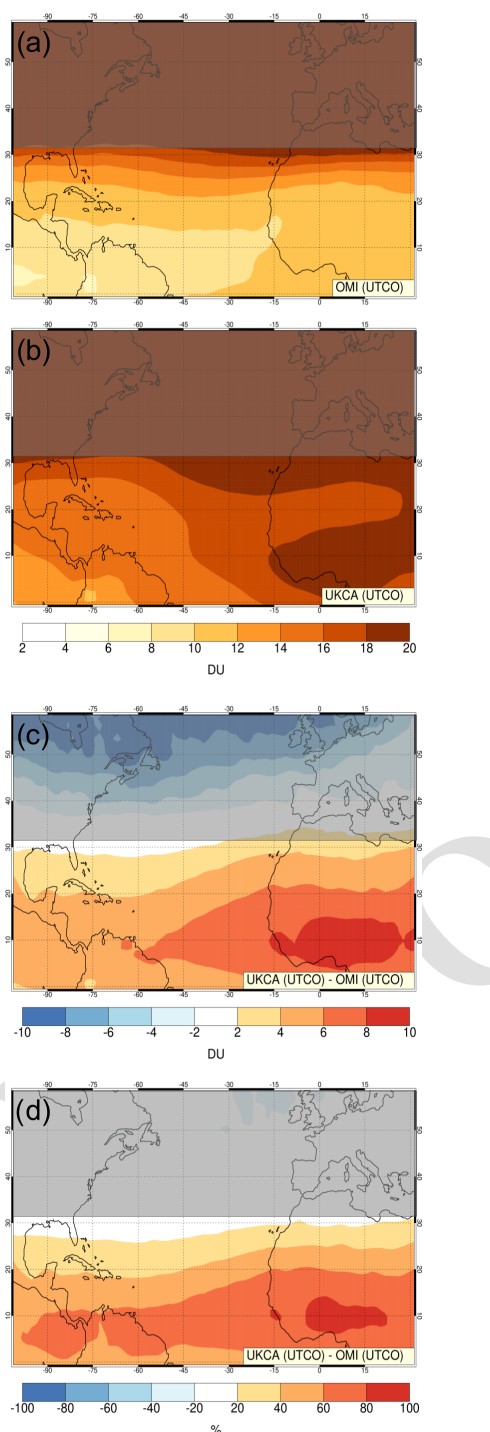

**Figure 3.** Upper tropospheric ozone column (450–170 hPa) multiannual means for the period 2005–2018: **(a)** OMI UTCO, **(b)** UKCA UTCO. **(c)** Difference between modelled and observed UTCO (DU). Panel **(d)** is the same as **(c)** but expressed as a percentage difference. Latitudes outside the tropics are shaded since this ozone subcolumn at mid-latitudes samples the lower stratosphere as well as the upper troposphere.

Isoprene emissions are largest in equatorial Africa, South America and South-east Asia; in UKCA these emissions are produced interactively and are diagnosed from the gross primary productivity of the terrestrial vegetation (Arneth et al., 2007; Pacifico et al., 2011). Present-day global isoprene emissions range from $\sim 500$ to $750\,\mathrm{Tg\,yr^{-1}}$ (Achakulwisut et al., 2015). The way isoprene affects ozone concentrations throughout the troposphere is complex and several studies have focused on the effect of isoprene emissions on tropospheric ozone (Paulot et al., 2012; Squire et al., 2015; Hollaway et al., 2017; Weber et al., 2021). Hollaway et al. (2017) showed that a reduction of $\sim 100\,\mathrm{Tg}$ in isoprene emissions results in a reduction in zonal mean ozone concentrations, with the largest difference ($\sim 2\,\mathrm{ppbv}$) in the tropical upper troposphere. However, the relative impact of isoprene on ozone is highly dependent on the chemical scheme used to describe isoprene oxidation and reactivity (Squire et al., 2015; Weber et al., 2021). Further uncertainty in the modelled seasonality and geographical distribution of isoprene emissions could also lead to a discrepancy between modelled and observed ozone.

Another possible source of model uncertainty is $NO_x$ emissions from lightning ($LiNO_x$). Lightning flashes are most frequent in the tropics and are linked to deep convective storms over Africa, South America and South-east Asia. $LiNO_x$ emissions in UKCA are parameterized (based on Price and Rind, 1992) and released partly at the surface but mostly within the depth of convective clouds; as a result, $LiNO_x$ emissions are largest in the tropical upper troposphere. The estimated range of $NO_x$ emissions from lightning varies between 2 and $8\,\mathrm{Tg\,(N)\,yr^{-1}}$ (Schumann and Huntrieser, 2007; Wang et al., 2013), 4 and $8\,\mathrm{Tg\,(N)\,yr^{-1}}$ (Martin et al., 2007), and more recent estimates of $9\,\mathrm{Tg\,(N)\,yr^{-1}}$ (Nault et al., 2017). Although $NO_x$ from lightning represents a small fraction of total $NO_x$ emissions, with $\sim 20.5\,\mathrm{Tg\,(N)\,yr^{-1}}$ coming from anthropogenic sources and $\sim 5.5\,\mathrm{Tg\,(N)\,yr^{-1}}$ from biomass burning (Lamarque et al., 2010), it has a disproportionately large influence on the tropospheric ozone budget. This is because ozone production efficiencies per unit $NO_x$ are an order of magnitude higher in the tropical upper troposphere (Pickering et al., 1990). It has been estimated that a reduction in $LiNO_x$ from 5 to $2.5\,\mathrm{Tg\,(N)\,yr^{-1}}$ results in a significant reduction, up to 40 %–60 %, in upper tropospheric ozone in the tropics (Liaskos et al., 2015). Lelieveld and Dentener (2000) showed that the contribution of lighting to ozone in the tropical upper troposphere is $\sim 50\%$, while Grewe (2007) used tagged tracers from different $NO_x$ sources and found lightning contributed 70 % of $NO_y$ and 40 % of ozone in the tropical upper troposphere. $LiNO_x$ parameterization is therefore a potentially large source of uncertainty in UKCA tropospheric ozone budget.

To investigate the source of UKCA tropical ozone bias, we look at the temporal evolution of modelled TCO and modelled ozone precursor emissions over the period of the model

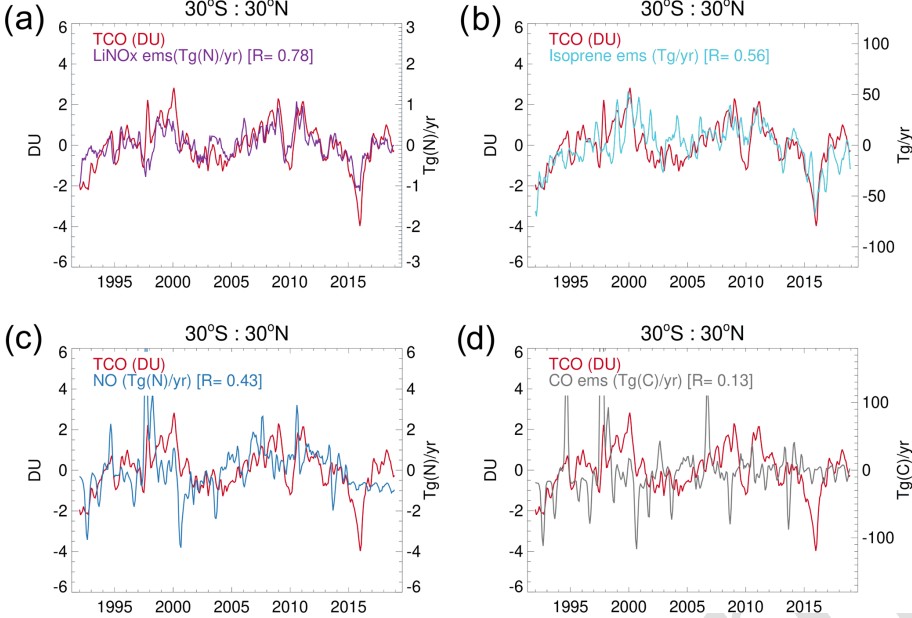

**Figure 4.** Deseasonalized and detrended time series anomalies showing Pearson's correlation coefficient ($R$) calculated between time series of modelled TCO and emissions of ozone precursors, averaged over the tropics, for the period 1992–2018.

integration (1992–2018). One way to understand how modelled TCO changes in response to changes in modelled emissions is to calculate the Pearson's correlation coefficient, $R$, and the coefficient of determination, $R^2$, between the time series of ozone and specific ozone precursor emissions.

Figure 4 shows deseasonalized and detrended time series anomalies averaged over the tropical region for 1992–2018. In UKCA, tropical TCO is strongly correlated ($R = 0.78$) to LiNO$_x$ emissions (Fig. 4a), which explain $\sim 60\%$ of the modelled ozone interannual variability ($R^2 = 0.61$); in contrast, surface NO$_x$ emissions (Fig. 4c) have a relatively lower correlation to TCO ($R = 0.43$), and about $\sim 20\%$ of tropical TCO variability can be attributed to changes in surface NO$_x$ ($R^2 = 0.18$). Amongst possible sources of VOC in the tropics, isoprene emissions (Fig. 4b) have a higher correlation to TCO than CO emissions (Fig. 4d). This suggests that the main production of ozone in the tropical troposphere occurs via reaction of LiNO$_x$ with isoprene and other long-lived hydrocarbons and is less sensitive to surface NO$_x$ emissions and CO emissions. As mentioned before, both LiNO$_x$ and isoprene emissions are calculated interactively in UKCA, therefore, given the high correlation with modelled ozone in the tropics, errors in the geographical distribution, seasonality or magnitude of such emissions will play a key role in how accurately tropical TCO is represented in the model. The sensitivity of modelled TCO to changes in the magnitude of LiNO$_x$ and isoprene emissions was further investigated through a number of sensitivity experiments (see Fig. S5), whereby global isoprene and LiNO$_x$ emission values were modified without changing their geographical or seasonal distributions. These confirmed that changes in LiNO$_x$ emis-

sions have a large impact on tropical TCO: in agreement with previous studies (Liaskos et al., 2015), a reduction in LiNO$_x$ emissions of $\sim 5$ Tg (N) results in a marked decrease in modelled TCO over large regions in the tropics (SF 5b), with the largest difference $\sim 10$–$15$ ppbv in the free troposphere, between 5 and 13 km, and smaller differences (2–5 ppbv) in the lower troposphere (SF 5a). Reducing isoprene emissions from $\sim 800$ to $\sim 260$ Tg yr$^{-1}$ (SF 5c and d) has a somewhat smaller impact on tropical ozone, and changes are vertically more homogeneous ($\sim 2$–$5$ ppbv throughout the depth of the troposphere). In this context, it is worth mentioning that a more complete chemistry scheme, which includes isoprene peroxy radical H shifts and HO$_x$ recycling, has been developed and tested in UKCA (Weber et al., 2021) and could lead to a different sensitivity of modelled tropospheric ozone to isoprene emissions.

Both temporal correlation analysis (Fig. 4) and sensitivity experiments (SF 5) point to LiNO$_x$ emissions as a major source of modelled TCO bias in the tropical upper troposphere.

## 3.3 Observed vs. modelled lightning flash rate

In order to understand where the model inaccuracy in the LiNO$_x$ emissions stems from, we compare observed and modelled climatology of lightning flashes for 1996–2013 (chosen as the period for which LIS-OTD flash rate observations are available). Figure 5 shows multiannual mean maps of observed (a) and modelled (b) lightning flash rates, while Fig. S6 shows the same comparison but using seasonally averaged maps. This analysis shows that (i) the model under-

estimates lightning flashes globally (with $7.8 \times 10^{10}$ flashes per year on average between 1996 and 2013, compared to $15.3 \times 10^{10}$ from LIS-OTD), (ii) the model underestimates the seasonal increase in lightning activity for the summer hemisphere, (iii) the model overestimates flashes in the tropics compared to mid-latitudes, and (iv) the model overestimates flashes over land compared to the ocean. Although the latter point has been addressed by a recent modification in the UKCA lightning flashes parameterization (Luhar et al., 2021), the modelled overestimate of flashes in the tropics and underestimate at mid-latitudes is still an issue (see Fig. 4 in Luhar et al., 2020). These model shortfalls in the geographical distribution and seasonality of lightning flashes are interlinked and mainly stem from the use of cloud diagnostics from the convection parameterization scheme being used to estimate cloud thickness in the lightning scheme. This is because deep convection in the model is nearly totally restricted to tropical regions resulting in a large fraction of extra-tropical convective storms and their seasonal shift to higher latitudes being underestimated, leading to an imbalance between tropical and extra-tropical lightning flashes. In fact, a lightning flash parameterization based on cloud ice flux (Finney et al., 2014) was shown to have a much better zonal distribution of flashes compared to observations. Another issue in this lightning $NO_x$ parameterization is that, in order to ensure the modelled global $LiNO_x$ emissions fall within the estimated range, a scaling factor is used to prescribe the amount of $LiNO_x$ emitted per flash. As a result, a relatively small number of modelled flashes, mostly concentrated in the tropics, will yield the entirety of the global emissions of $LiNO_x$, leading to an overestimate in $LiNO_x$ and consequent widespread ozone bias in the tropical upper troposphere. The model underestimate of flashes and $LiNO_x$ emissions outside the tropics could also potentially contribute to an underestimate of tropospheric ozone at mid-latitudes.

We now more specifically address the way that errors in modelled $LiNO_x$ emissions affect modelled tropospheric ozone in the North Atlantic. Similarly to our approach in Fig. 4a, we investigate how modelled TCO, this time in the tropical North Atlantic, correlates to $LiNO_x$ emissions from a number of source regions and seasons. Note that due to the strongly seasonal nature of regional $LiNO_x$ emissions (see also Fig. S6), correlation coefficients are calculated only for the seasons with the largest regional emissions (JJA for East Asia and SON for the tropical North Atlantic). This analysis (summarized in Fig. 5c) reveals that TCO in the tropical North Atlantic has a sizeable correlation with local $LiNO_x$ emissions (with the largest $R_{SON}$ value of 0.68 in the autumn) but an even stronger correlation with $LiNO_x$ emissions from South-east Asia (defined as the region delimited by the box in Fig. 5a and 5b) in the summer ($R_{JJA} = 0.82$), explaining $\sim 70\%$ of its summer variability ($R_{JJA}^2 = 0.67$). This is consistent with modelled ozone, produced from $LiNO_x$ over East Asia in summer, being efficiently transported to

the tropical North Atlantic by the prevailing north-easterly trade winds. Over Asia, modelled lightning flashes have a pronounced summer maximum (Fig. 5d), whilst observed lightning flashes display a broader maximum across spring and summer (see Figs. 5d and S6). This discrepancy in the seasonality of $LiNO_x$ emissions over Asia could be responsible for modelled TCO in the tropical North Atlantic having a stronger summer maximum compared to observations (Fig. 2e).

## 4   Time evolution of tropospheric ozone: interannual and decadal variability

We now investigate how tropospheric ozone in the North Atlantic has been changing over time, with a focus on the recent past (2005–2018). Tropospheric ozone has a large seasonal cycle, particularly at mid-latitudes (as shown in Fig. 2), therefore in order to investigate the interannual and decadal variability of tropospheric ozone in the North Atlantic the seasonal signal has been removed from the observed and modelled ozone data. In a recent paper, Archibald et al. (2020a) have shown that modelled tropospheric ozone has been increasing sharply throughout the early 1900s and then more slowly from the 1960s to present day. They postulate that this is consistent with a strong increase in anthropogenic ozone precursors emissions during most of the 1900s and showed that tropospheric ozone chemical production peaked in the 1990s and has been slowly decreasing after that, as anthropogenic $NO_x$ emissions have been levelling off. This suggests that emissions from non-anthropogenic sources and transport of ozone from the stratosphere are likely to play a major role on the interannual and decadal variability of tropospheric ozone from 2000s onward. In this section we aim to investigate recent interannual and decadal variability of tropospheric ozone in the North Atlantic and the drivers of such variability.

### 4.1   Observed vs. modelled tropospheric ozone interannual variability and trends in the North Atlantic

Figure 6 shows a comparison between observed and modelled time evolution of TCO (first column) and LTCO (second column); these are plotted as deseasonalized time series anomalies, averaged over the North Atlantic (top row), mid-latitude North Atlantic (middle row) and tropical North Atlantic (bottom row). The observed time series indicate ozone remained constant or slightly decreased between 2005 and 2010, followed by an increase and, in the case of OMI LTCO, a further decrease from 2014–2016 onwards. This highlights that ozone changes in the North Atlantic are not constant over time, and therefore trends can be very different depending on the period used to calculate them. Several different approaches have been used in the literature to estimate ozone trends, ranging from fairly complex, such as "multivariate

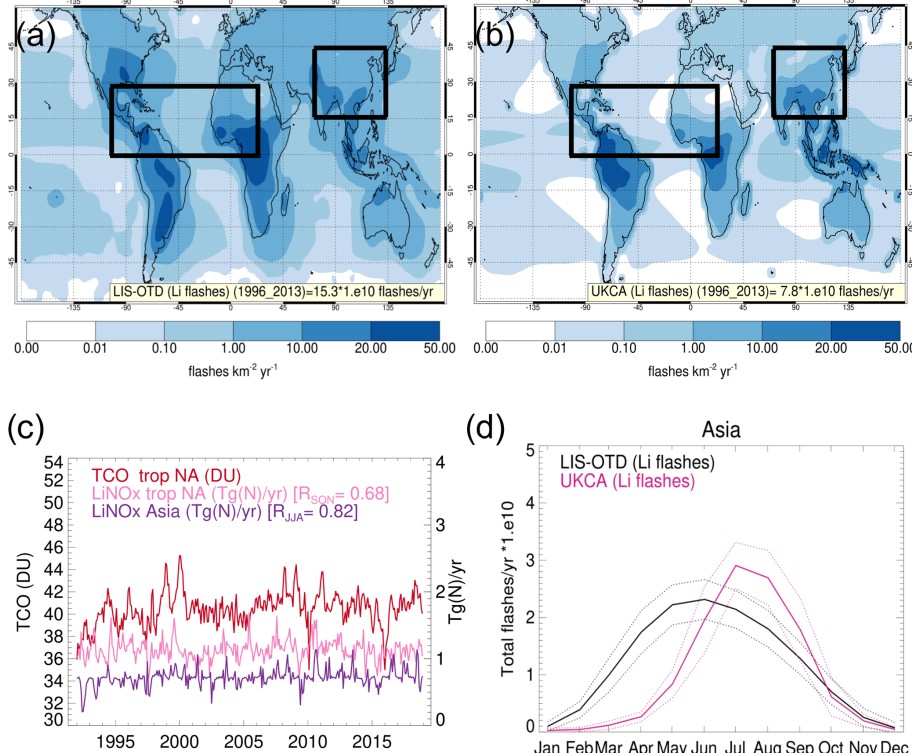

**Figure 5.** Comparison of observed **(a)** and modelled **(b)** multiannual mean maps of lightning flash rates for 1996–2013; black boxes indicate tropical North Atlantic and Asia. Panel **(c)** shows deseasonalized and detrended time series of TCO in the tropical North Atlantic and regional $LiNO_x$ emissions: large seasonal correlations were found for tropical North Atlantic TCO and $LiNO_x$ from tropical North Atlantic and Asia ($R$ values are shown in the legend for seasons with largest correlation). Panel **(d)** shows observed and modelled seasonal cycles of lightning flashes for the period 1996–2013 over Asia.

linear regression" (Ziemke et al., 1997, 2019) to simple linear regression fit (e.g. Gaudel et al., 2018). In this work we chose a simple linear regression method since our aim is to compare differences between modelled and observed trends, rather than provide the most accurate estimate of ozone trends between 2005 and 2018. The linear trends and errors over this period are noted in the legends in Fig. 6: both satellite datasets show positive values in the tropical part of the domain (1.6–1.7 DU per decade, equivalent to ∼ 5 % per decade for TCO and ∼ 10 % per decade for LTCO). These are relatively large trends, which are comparable in magnitude to modelled trends for the early part of the 20th century, when anthropogenic emissions of ozone precursors were increasing significantly.

While trends for the two observed datasets agree well over the tropical part of the domain, OMI-MLS trends are larger in the mid-latitude North Atlantic compared to OMI, which shows a zero trend within the estimated error. These findings are consistent with previous studies (Gaudel et al., 2018), which found generally positive ozone trends in the recent past (over a period ranging between 1996 and 2016, depending on the specific datasets) and a large spread in trends estimated from different satellite platforms. With the exception of two satellite datasets, whose record started in 2008 and showed small negative trends, Gaudel et al. (2018) found positive linear trends in TCO for most other datasets over the Northern Hemisphere to be between 1.7 and 2.7 DU per decade. In contrast to observed trends, UKCA ozone trends tend to be much smaller in magnitude and effectively zero (within the error) for both the tropical and mid-latitude part of the domain. Skeie et al. (2020) investigated ozone trends in CMIP6 model simulations; while it is clear that modelled ozone has increased significantly between 1850 and 2010, it is hard to pinpoint the sign and magnitude of the ozone trends from CMIP6 models in the more recent decades. Figure 2 in Skeie et al. (2020) shows that observed ozone trends for 2000–2010 are less than 5 % per decade, while modelled trends for the same period show many models have trends very close to zero, and generally within ± 2 % per decade. Although the period we are investigating (2005–2018) is not the same as shown in Skeie et al. (2020), their findings suggest that UKCA ozone trends in the more recent decades are comparable to other CMIP6 models and that accurately estimating tropospheric ozone trends over relatively short time periods remains a challenge due to ozone's large interannual variability. Figure 6 can also give an insight into the interan-

nual variability of tropospheric ozone: the temporal correlation between observed and modelled time series in Fig. 6 is generally low (ranging between 0 and 0.25 for the different domains and subcolumns), suggesting that the factors driving interannual variability might be different for observed and modelled tropospheric ozone (see Sect. 4.2 and 4.3 for further discussion on drivers of North Atlantic ozone interannual variability).

Another way to compare differences in the geographical distribution of observed and modelled trends is through the use of linear trend maps; Fig. 7 shows observed (top row) and modelled (bottom row) tropospheric ozone trends for TCO (first column) and LTCO (second column); the same information for the global scale is provided in Fig. S7. Despite differences in the methods used to calculate trends, the observed TCO values are consistent with previous studies (Ziemke et al., 2019; Gaudel et al., 2018). Although OMI data were previously used to calculate TCO trends by extrapolating LTCO to TCO, assuming volume mixing ratios in the lower troposphere to apply up to the tropopause (Gaudel et al., 2018), this is the first time that OMI data were used to calculate trends in LTCO. Since changes in tropospheric ozone in the past couple of decades are small relative to its interannual variability (Archibald et al., 2020a; Gaudel et al., 2018), accurately estimating such trends is a challenge from both a modelling and observational perspective and continuing remote sensing observations programmes are therefore vital to provide a long-term observational record and allow for more accurate ozone trend estimates.

## 4.2 Drivers of North Atlantic ozone interannual and decadal variability in UKCA

In this section we address what drives interannual and decadal variability of ozone in UKCA, with a view to understanding the reasons for the discrepancies between observed and modelled tropospheric ozone trends and interannual variability in the North Atlantic. In Fig. 4 we used deseasonalized and detrended time series anomalies to show that modelled interannual variability of tropospheric ozone in the tropics is largely driven by changes in $LiNO_x$ emissions. Here we perform a similar analysis for the North Atlantic region. We calculate correlation coefficients between deseasonalized and detrended time series of TCO in the North Atlantic (for 1992–2018) and various possible ozone drivers including ozone precursors emissions and ozone transported from the stratosphere (O3S tracer). The correlation coefficients are calculated between TCO in the North Atlantic and ozone drivers from three different regions (the North Atlantic, the Northern Hemisphere and the tropics), which allows us to distinguish between local and non-local drivers of North Atlantic ozone; results are summarized in Table 1. For values of $R$ greater than 0.7, the coefficient of determination $R^2$ is greater than 50 % (meaning that 50 % of the ozone variability can be attributed to that specific ozone driver);

therefore, correlation coefficients in the table greater than 0.7 are highlighted in bold. Table 1 shows that local changes in ozone transported from the stratosphere dominate the modelled IAV of tropospheric ozone in the North Atlantic, with tropical emissions of $LiNO_x$ also playing a very important role. Please note that because the different sources of ozone are not independent of each other and might themselves be correlated, the $R^2$ values from different ozone sources do not add up to one. Time series of key drivers and their time correlations are also shown in Fig. S8.

Having identified ozone from STT and tropical $LiNO_x$ emissions as the main drivers of modelled North Atlantic ozone interannual variability, we investigate whether decadal changes in such drivers could also be responsible for modelled ozone decadal variability and trends. In order to do that, we analyse modelled trends in ozone drivers and try to assess if ozone from STT and tropical $LiNO_x$ emissions have trends that are consistent with TCO trends. For this analysis, we divide our model integration (1992–2018) into three separate periods characterized by positive, negative and zero trends in North Atlantic TCO (Fig. 8), and we calculate trends in modelled TCO, ozone precursors emissions and ozone transported from the stratosphere in each of the sub-periods, which are summarized in Table 2. In the first period, all ozone drivers in the model have positive trends, leading to the strong positive trend in modelled TCO. In the second period, tropical emissions of $LiNO_x$ have a zero trend and O3S has a negative trend, which results in a negative trend in modelled TCO. In the third period $LiNO_x$ and O3S have zero or very small trends, resulting in a zero trend in modelled TCO, despite a negative trend in tropical isoprene emissions. This suggests that modelled trends in O3S and lightning $NO_x$ emissions could be driving decadal variability in modelled North Atlantic TCO, as trends in these key drivers are consistent with modelled ozone trends for three different periods in the model simulation. Although this analysis focuses on drivers of North Atlantic ozone interannual and decadal variability, results from Table 2 and Fig. 8 suggest that tropical emissions of lightning $NO_x$ and ozone transported from the stratosphere have a similarly large influence on modelled ozone trends in the Northern Hemisphere and the tropics.

We now investigate reasons for the discrepancy between observed and modelled trends in the North Atlantic (for 2005–2018) and why the model might underestimate the observed trends. Whilst there is no clear evidence (from observed lightning flash rates) that lightning activity has increased in the recent past (Cecil et al., 2014; Kaplan and Lau, 2021), the model could be underestimating the trend in ozone transported from the stratosphere, leading to an underestimate of North Atlantic TCO trend compared to observations. Changes in ozone transported from the stratosphere could result from (a) changes in lower stratospheric ozone concentrations or (b) changes in the downward transport from the stratosphere. The latter is due to changes in the strength of the Brewer Dobson circulation (Butchart, 2014) which is in-

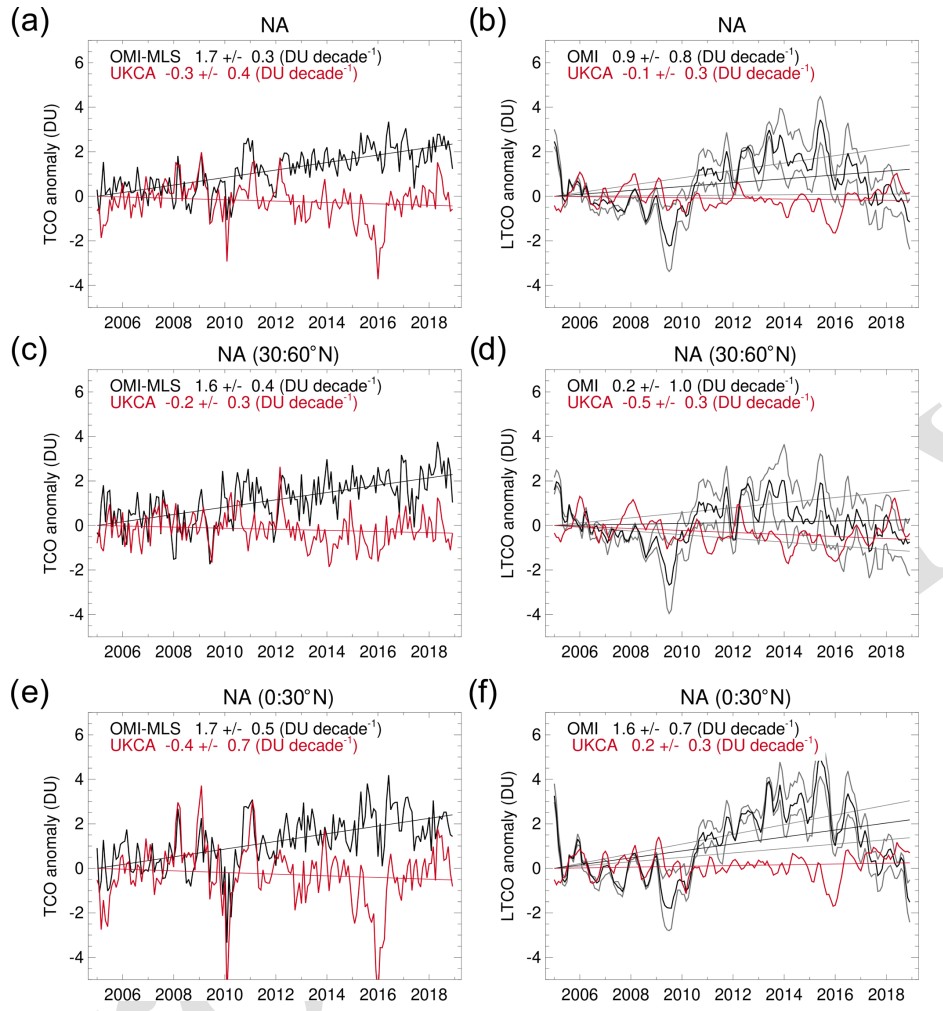

**Figure 6.** Tropospheric ozone deseasonalized time series over the North Atlantic (**a, b**), mid-latitude North Atlantic (**c, d**) and tropical North Atlantic (**e, f**). The first column shows TCO, and the second column shows LTCO. Trends were calculated using a least-squares linear regression method on monthly deseasonalized time series. Grey lines in panels (**b**), (**d**) and (**f**) represent uncertainties associated with changes in OMI cross-track sampling. This uncertainty is largest at northern mid-latitudes and increases over time.

**Table 1.** Correlation coefficients between modelled TCO in the North Atlantic and ozone drivers in three different regions (the North Atlantic, the Northern Hemisphere and the tropics). Correlation coefficients higher than or equal to 0.7 are highlighted in bold. Correlation coefficients higher or equal to 0.7 are highlighted in bold. All $R$ values, except for North Atlantic isoprene emissions, are significant to the 95 % confidence interval.

| SOURCE | $R$ between North Atlantic TCO and source (1992–2018) | | |
|---|---|---|---|
| | NA | NH | Tropics |
| LiNO$_x$ emissions | 0.15 | 0.40 | **0.72** |
| Isoprene emissions | 0.01 | 0.13 | 0.50 |
| Surf NO$_x$ emissions | −0.12 | 0.39 | 0.35 |
| Ozone from STT | **0.77** | **0.76** | 0.62 |

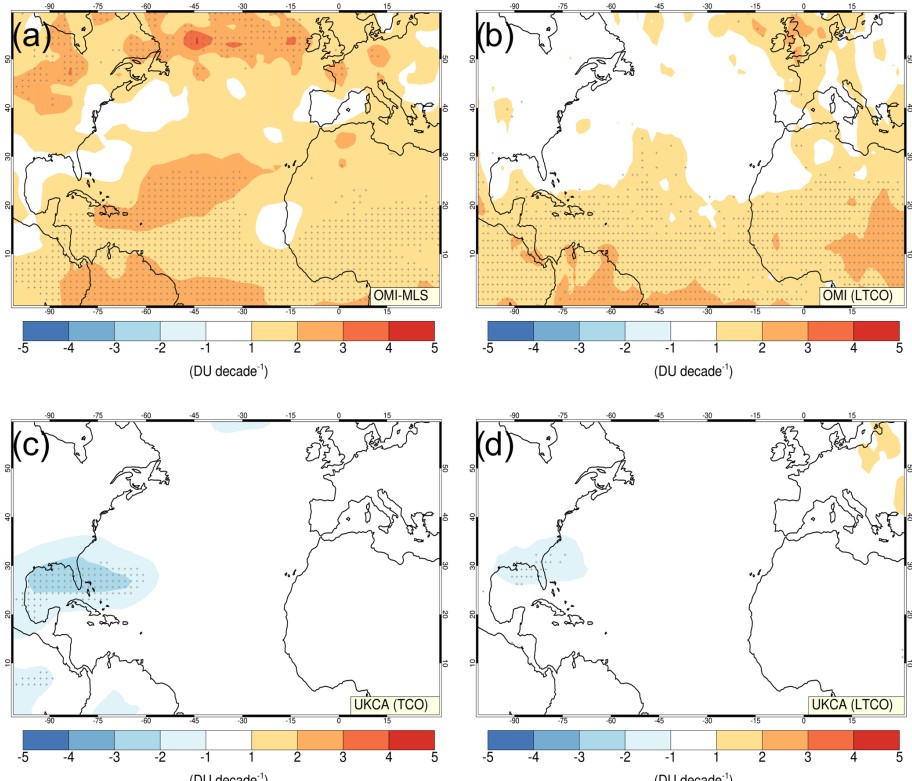

**Figure 7.** Tropospheric ozone trends for the period 2005–2018: OMI-MLS TCO trend **(a)**, OMI LTCO trend **(b)**, UKCA TCO trend **(c)** and UKCA LTCO trend **(d)**. The stippling indicates where trends are significant to the 95 % confidence level, based on the standard error of the residuals.

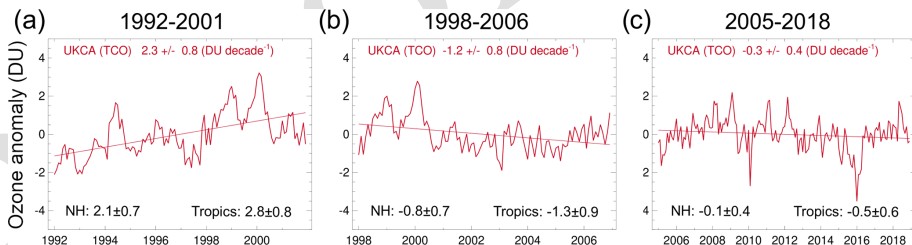

**Figure 8.** Modelled tropospheric ozone time series in the North Atlantic and linear trends for three different periods. Additional trends for TCO in the Northern Hemisphere and the tropics are noted at the bottom of each panel.

trinsically hard to evaluate as it is poorly constrained by observations. Analysis of a O3S-C model tracer (which measures modelled downward transport) shows that there has been no significant change in modelled downward transport between 2005 and 2018. We therefore investigated changes in lower stratospheric ozone concentrations using the OMI data sampled between 170 and 50 hPa ($\sim$ 13–20 km) and the equivalent UKCA data. We specifically focus on trends in modelled and observed lower stratospheric ozone in spring and autumn and around 30° N and 30° S, which are the times and locations associated with the largest STT (Škerlak et al., 2014; Williams et al., 2019). Figure 9 shows seasonal trend maps for OMI and UKCA columns, sampled between 170

and 50 hPa: the black boxes highlight the regions and seasons most relevant for STT, whilst shaded areas indicate regions where the vertical subcolumn samples the upper troposphere as well as the lower stratosphere. From the analysis of Fig. 9 we can see a clear positive trend in observed lower stratospheric ozone in the mid-latitude North Atlantic in spring that is not mirrored in the model. An additional trend analysis, using MLS lower stratospheric ozone data for 2005–2018 (not shown), is consistent with the positive OMI trends in the black boxes in Fig. 9. This suggests that a lack of positive trend in spring and autumn modelled lower stratospheric ozone at mid-latitudes might be responsible for the

**Table 2.** Trends of modelled TCO and ozone drivers over different periods and regions. Panel **(a)** shows absolute trend values and associated error of the trend estimate (in brackets). Panel **(b)** shows percentage change per decade relative to the concentration of each species at the beginning of the period in question (or zero if the trend is smaller than the error of the trend estimates). Trends for species or regions having a strong correlation ($R > 0.7$) to North Atlantic TCO (see Table 1) are highlighted in bold.

| **(a)** | 1992–2001 TREND | | | 1998–2006 TREND | | | 2005–2018 TREND | | |
|---|---|---|---|---|---|---|---|---|---|
| | NA | NH | Tropics | NA | NH | Tropics | NA | NH | Tropics |
| UKCA TCO DU per decade | 2.3 (0.8) | 2.1 (0.7) | 2.8 (0.8) | −1.2 (0.8) | −0.8 (0.7) | −1.3 (0.9) | −0.3 (0.4) | −0.1 (0.4) | −0.5 (0.6) |
| LiNO$_x$ Tg (N) per decade | −0.1 (0.1) | 0.2 (0.2) | **0.8 (0.3)** | 0.0 (0.1) | −0.1 (0.2) | **0.0 (0.3)** | −0.1 (0.1) | 0.0 (0.1) | **−0.1(0.2)** |
| Isoprene Tg per decade | −7 (6) | −3 (10) | 61 (14) | −3 (7) | −4 (14) | −13 (19) | −8 (3) | −15 (5) | −31 (10) |
| Surf NO$_x$ Tg (N) per decade | 0.2 (0.4) | 1.7 (0.7) | 4.1 (1.6) | −1.7 (0.5) | 4.5 (1.3) | 3.8 (1.4) | −2.5 (0.2) | 0.2 (0.4) | 2.3 (0.5) |
| Ozone from STT DU per decade | **0.9 (0.5)** | **0.9 (0.4)** | 0.7 (0.4) | **−0.5(0.4)** | **−0.6(0.3)** | −0.9 (0.3) | **0.1 (0.3)** | **0.0 (0.2)** | −0.2 (0.2) |

| **(b)** | 1992–2001 TREND % | | | 1998–2006 TREND % | | | 2005–2018 TREND % | | |
|---|---|---|---|---|---|---|---|---|---|
| | NA | NH | Tropics | NA | NH | Tropics | NA | NH | Tropics |
| UKCA TCO | 7 | 6 | 8 | −3 | −2 | −3 | 0 | 0 | 0 |
| LiNO$_x$ emissions | 0 | 0 | **17** | 0 | 0 | **0** | 0 | 0 | **0** |
| Isoprene emissions | −5 | 0 | 14 | 0 | 0 | 0 | −10 | −8 | −9 |
| Surf NO$_x$ emissions | 0 | 4 | 20 | −10 | 10 | 15 | −25 | 0 | 10 |
| Ozone from STT | **6** | **6** | 5 | **−3** | **−3** | −6 | **0** | **0** | 0 |

model underestimating the observed tropospheric ozone positive trends between 2005 and 2018.

This is a somewhat surprising result, as previous studies (Ball et al., 2020; Dietmüller et al., 2021) suggest that models have larger trends in lower stratospheric ozone at mid-latitude compared to observations from a merged stratospheric ozone dataset (Ball et al., 2018). However, trends in those studies were calculated from an earlier period (1998–2018) and for slightly different vertical columns, so there are a number of possible reasons why our results differ. The main question is why UKCA is underestimating spring trends in lower stratospheric ozone compared to similarly sampled OMI data. Positive ozone trends in the upper stratosphere have been observed since 1998 (e.g. Harris et al., 2015; Steinbrecht et al., 2018; Weber et al., 2018; Ball et al., 2018), which can be attributed to the reduction in ozone-depleting substances driven by the Montreal protocol (WMO, 2018) and the cooling of the stratosphere due to increase in greenhouse gases and consequent decrease in the rate of ozone destruction (Portmann and Solomon, 2007). However, ozone in the mid-latitude lower stratosphere (between 170–50 hPa) is more directly influenced by the lower branch of the Brewer–Dobson circulation (Abalos et al., 2014; Dietmüller et al., 2018) and therefore more strongly linked to lower stratospheric ozone in the tropics (Dietmüller et al., 2021). Because of this, ozone in the mid-latitude lower stratosphere is more likely to be affected by changes in the strength of tropical upwelling, which can be modulated by ENSO and the quasi-biennial oscillation (QBO) (Neu et al., 2014), and driven by greenhouse gas increases and global warming (Butchart, 2014).

One possibility is that UKCA might overestimate the trend in tropical upwelling between 2005–2018, leading to an increased transport of ozone-poor air from the tropical troposphere to the mid-latitude lower stratosphere, and this could in turn result in an underestimate in ozone trends in the mid-latitude lower stratosphere. This grants further investigations that are, however, outside the scope of this study.

## 4.3 Dynamical response of modelled STT and LiNO$_x$ emissions and their impact on North Atlantic ozone IAV

In Sect. 4.2 we have identified tropical NO$_x$ emissions and transport of ozone from the stratosphere as the two main drivers of modelled North Atlantic ozone IAV. In this section we investigate the impact of external dynamical forcing on the IAV of observed and modelled North Atlantic ozone, and we further analyse possible reasons for the discrepancy between observed and modelled ozone IAV. Unlike anthropogenic emissions and other natural ozone precursor emissions (such as soil NO$_x$ and biomass burning), lightning NO$_x$ and ozone transported from the stratosphere are not imposed as model input but are diagnosed interactively within the model. Because of this, such sources of tropospheric ozone are therefore influenced by dynamical processes and climate modes of variability. ENSO affects convection and can therefore impact the strength and geographical distribution of lightning and LiNO$_x$ emissions (Chronis et al., 2008; Sátori et al., 2009), whilst the AO and NAO affect the strength and location of the storm track in the North Atlantic basin and can therefore impact downward transport from the stratosphere (Lamarque and Hess, 2004; Pausata et al., 2012; Reutter et al., 2015).

In this study, we have analysed correlation maps between observed and modelled ozone and the AO and NAO indices for different vertical subcolumns and different seasons. The maps were derived by correlating monthly detrended ozone time series for each grid point to the AO and NAO index. Very similar correlation patterns were found when

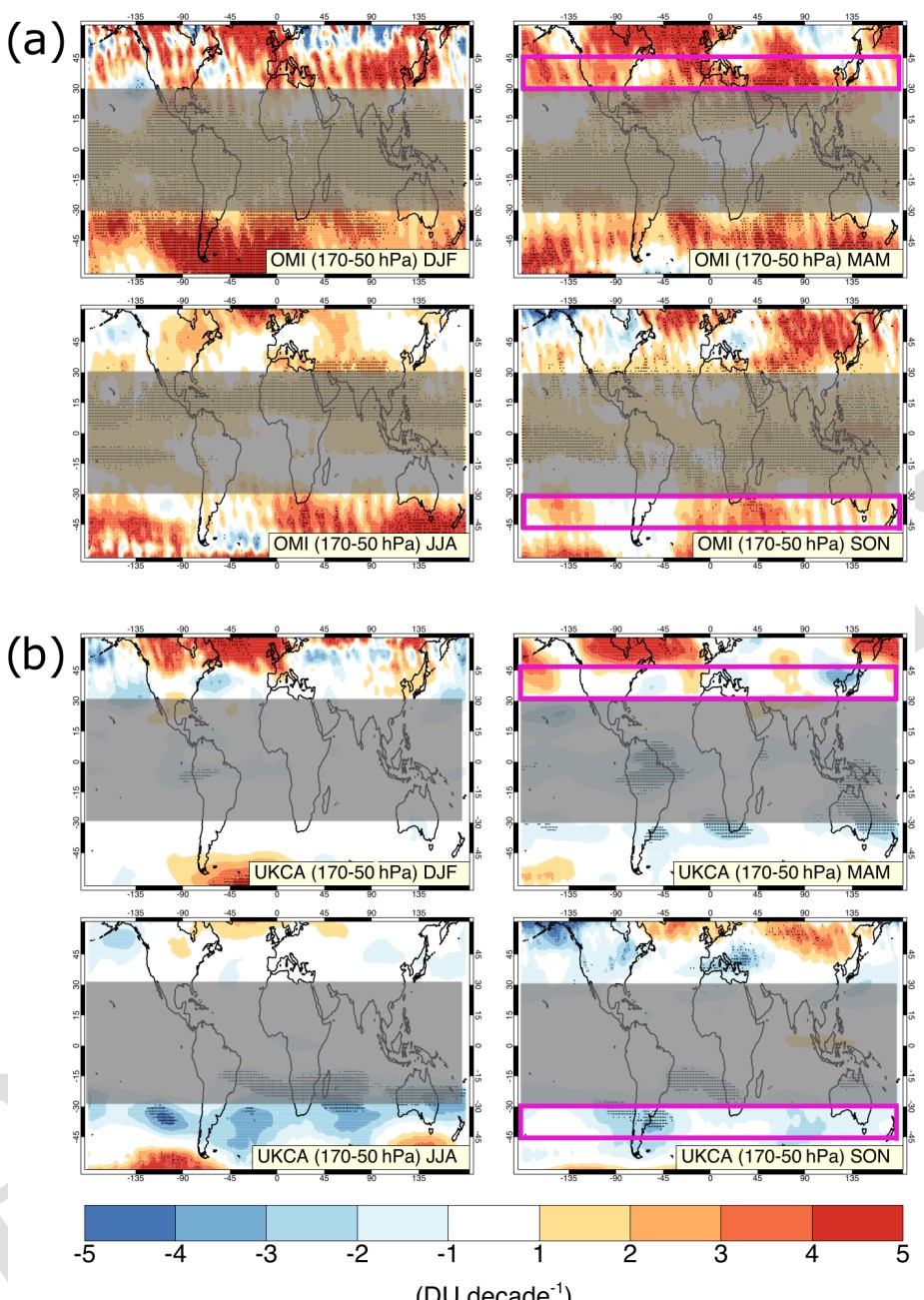

**Figure 9.** Seasonal trend maps of ozone columns sampled between 170–50 hPa for OMI **(a)** and UKCA **(b)**. The black boxes highlight regions and seasons most relevant for STT; shaded areas indicate regions where the vertical subcolumn samples the upper troposphere as well as the lower stratosphere.

using AO and NAO indices, and strongest correlation values were observed between ozone and the AO in the winter months (DJF); correlation maps between ozone and AO index in DJF are therefore shown. In Fig. 10, observed ozone datasets (top row) show a consistent correlation pattern with the AO across all vertical subcolumns, i.e. negative correlation for latitudes between 30 and 50° N and positive correlation around 20° N, although correlation values are decidedly

smaller in the lower troposphere. Modelled ozone (middle row) shows a strikingly similar correlation pattern to OMI in the 450–170 hPa layer (Fig. 10c and f), but there are significant differences in the way modelled TCO and LTCO respond to forcing by the AO compared to observations. This suggests that the mechanism driving the ozone response to the AO is well captured by the model in the 450–170 hPa col-

umn but not in the lower troposphere, leading to differences in the full tropospheric column.

In order to investigate the role of STT on North Atlantic ozone response to the AO and NAO we used two modelled idealized tracers, O3S and O3S-C, which represent "ozone from STT" and its "downward transport from the stratosphere", respectively. The correlation maps between the AO and O3S-C are shown in the bottom row of Fig. 10 and are very similar to correlation maps between the AO and O3S (not shown); regions where observed ozone is increased or decreased in response to the AO coincide with regions where modelled downward transport of ozone-rich stratospheric air is increased or decreased in response to the AO. The remarkably similar patterns between the observed ozone correlation maps and the modelled STT tracer correlation maps offer additional evidence that the impact of the AO on STT plays a large role in the tropospheric ozone response to the AO and NAO in winter, as previously inferred by Lamarque and Hess (2004) and Pausata et al. (2012). Despite capturing the impact of the AO and NAO on STT, and therefore correctly replicating the ozone response in the 450–170 hPa region, modelled ozone in the troposphere and lower troposphere does not respond to the AO in the same way as observed ozone. This could be due to either long-range transport of ozone from other regions not being correctly represented or the fraction of ozone from STT reaching the lower troposphere being underestimated in the model.

ENSO is a leading mode of variability in the tropics driven by anomalies in sea surface temperature across the Pacific Ocean (Alexander et al., 2002). The geographical location of convection and associated lightning $NO_x$ emissions shifts with ENSO phases: positive ENSO favours convection over the central Pacific and Africa, while negative ENSO favours convection over South America and the Maritime Continent. The impact of ENSO on tropospheric and stratospheric ozone has been previously investigated (Ziemke et al., 2010; Oman et al., 2013; Tweedy et al., 2018; Olsen et al., 2019). Here we focus on the response of modelled North Atlantic ozone to ENSO and discrepancies between modelled and observed ozone response to ENSO. Figure 11 shows deseasonalized and detrended time series anomalies of (a) observed and modelled ozone in the North Atlantic and (b) tropical lightning flashes. Correlation coefficients between these time series and the ENSO3.4 index are reported in the legends; note that the lightning flash time series is plotted for a different period (1996–2013) due to the availability of lightning flash observations. Also note that both panels in Fig. 11 show time series of the negative ENSO index (−ENSO). From Fig. 11a it is clear that, whilst observed ozone datasets have no significant correlation to ENSO, modelled ozone is fairly strongly correlated to −ENSO, with the lower tropospheric column being somewhat less influenced by ENSO compared to the full tropospheric column. The reason for the different response of observed and modelled North Atlantic ozone to ENSO can be explained by Fig. 11b: observed

lightning flashes in the tropics are not significantly correlated to −ENSO, but modelled lightning flashes are correlated to −ENSO. The increased tropical lightning in the model, associated with negative ENSO events, leads to increased $LiNO_x$ emissions in the tropics, which drives an increase in modelled North Atlantic ozone. Analysis of the geographical distribution of lightning flashes suggests that the reason for the discrepancy between observed and modelled lightning flash responses to ENSO arises from the convection and lightning parameterizations. Oceanic convection is underestimated in the model, leading to an underestimate of $LiNO_x$ emissions and ozone production in the tropics during positive ENSO events.

Since modelled tropospheric ozone in the North Atlantic responds differently to AO and ENSO compared to observed ozone, the interannual variability of modelled tropospheric ozone does not match the observed ozone interannual variability. In contrast, temporal correlation between observed and modelled stratospheric ozone (measured between 170–50 hPa) is fairly high, with correlation coefficients as high as 0.75 for the mid-latitude North Atlantic; this does not occur for model simulations where nudging is not applied, suggesting that stratospheric ozone interannual variability is driven by dynamical processes whose temporal evolution is correctly captured in a nudged model integration.

## 5  Conclusions

This work aimed to investigate the seasonal, interannual and decadal variability of tropospheric ozone in satellite data and 3D CCM output from the UM-UKCA model and to understand what factors are driving such changes, with a specific focus on the North Atlantic region.

The model does a good overall job of capturing the ozone burden and spatial distribution of ozone in the North Atlantic. The seasonal cycle of ozone in the North Atlantic is well captured by the UKCA model. Both model and observations show a broad maximum in spring and early summer that is consistent with the two major sources of tropospheric ozone: transport from the stratosphere (with a maximum in late spring and early summer) and photochemical production from ozone precursors' emissions (with a maximum in summer). However, analysis of observed and modelled deseasonalized ozone time series for the period 2005–2018 shows that the interannual variability of ozone is different in the model compared to the observations. Whilst modelled LTCO has a smaller interannual variability compared to OMI, modelled TCO seems to have a larger interannual variability in the tropical North Atlantic. In order to understand the drivers of modelled ozone interannual variability in the North Atlantic we calculated correlation coefficients for the deseasonalized and detrended time series anomalies of ozone and a number of possible ozone drivers. The interannual variability of UKCA ozone in the North Atlantic is very strongly corre-

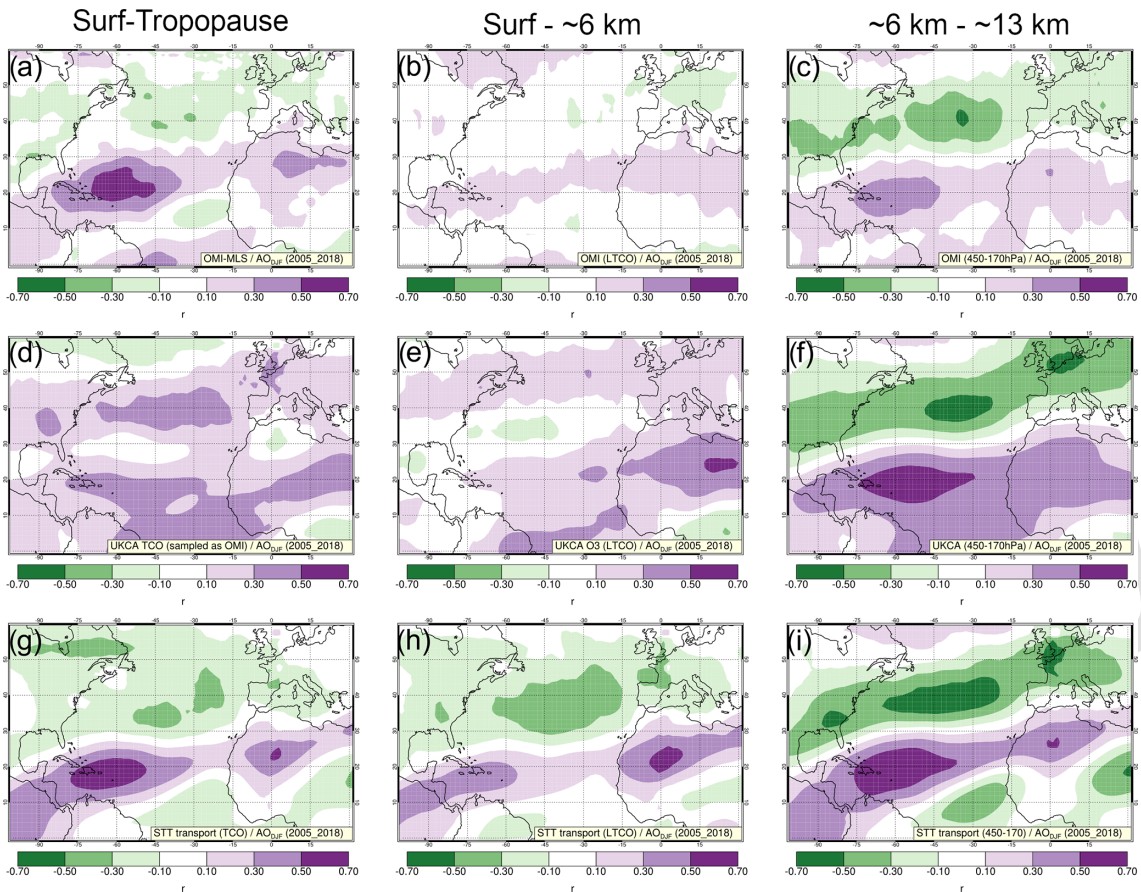

**Figure 10.** Correlation maps between the AO (in DJF) and observed ozone (**a, b, c**), modelled ozone (**d, e, f**) and modelled stratosphere to troposphere transport (**g, h, i**). The first column shows correlation for variables integrated over the full troposphere (surface to tropopause), while the second and third columns show correlation for variables integrated over the OMI subcolumns: surface–450 hPa ($\sim$ 0–6 km) and 450–170 hPa ($\sim$ 6–13 km).

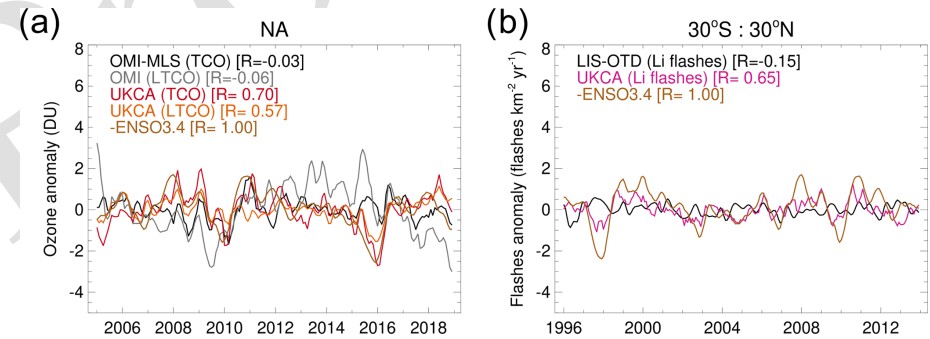

**Figure 11.** Deseasonalized and detrended time series anomalies of (**a**) observed and modelled ozone in the North Atlantic and (**b**) tropical lightning flashes. The −ENSO index time series is also plotted for comparison. Correlation coefficients ($R$) between each time series and the −ENSO index are noted in the legends.

lated to interannual variability of ozone transported from the stratosphere (quantified using a modelled idealized tracer) and tropical lightning $NO_x$ emissions. The interannual variability of ozone transported from the stratosphere and tropical lightning $NO_x$ emissions are in turn influenced by large-

scale modes of variability such as AO, NAO and ENSO, but the modelled response to such modes of variability is different from the observed response. In particular, tropical lightning and ozone are too strongly correlated to ENSO compared to observations. Using OMI data retrieved in discrete

vertical layers, in addition to model idealized tracers, we find that the dynamical response of STT to the AO can explain the observed tropospheric ozone response to the AO and NAO in winter.

Using OMI height-resolved data, in addition to OMI-MLS TCO, we can confirm that UKCA positive ozone bias in the tropics is the result of a large bias in the tropical upper troposphere and a much smaller bias in the tropical lower troposphere. The model tropical bias is larger than the difference in TCO values associated with a different choice of tropopause, and it is therefore not an artefact of tropospheric column definition.

Further analysis, including sensitivity experiments and comparison with an observed lightning flash climatology from LIS-OTD, shows that the positive bias in the tropical upper troposphere is due to shortfalls in the convection and lightning flash parameterizations in the model, leading to an overestimate of lightning $NO_x$ emissions and consequent overestimate of ozone formation in the tropics. Efforts are currently underway to improve the lightning parameterization in the model.

The model negative bias at southern mid-latitudes and high latitudes and northern high latitudes, relative to OMI-MLS TCO, is not supported by comparison with OMI data: analysis of UKCA and OMI ozone in the lower troposphere, with and without the use of satellite operator, suggests that the negative bias at such latitudes relative to OMI-MLS could be the result of incorrect model sampling due to the fact that averaging kernels are not available for the OMI-MLS dataset. This is further supported by generally good agreement between modelled and Bodecker Scientific ozone vertical profiles at mid-latitudes.

Both satellite datasets show a positive linear trend in ozone of $\sim 1.6$–$1.7\,\mathrm{DU}$ per decade for the tropical North Atlantic, but observed trends differ for mid-latitude North Atlantic. UKCA has a tendency to underestimate ozone trends both in the North Atlantic and globally. Our analysis points to differences between observed and modelled lower stratospheric ozone trends in spring as a possible source for the model's underestimate of tropospheric ozone trends. Estimating trends from both satellites and models is still a challenge since decadal changes are small compared to seasonal and interannual variability. However, improvements to the quality of data from satellite UV sounders, which extend back to 1995, and continued monitoring from current and planned satellites should enable shorter- and longer-term variability to be better discriminated. We encourage further studies that incorporate pairs of ozone tracers (e.g. O3S and O3S-C), as we have done in this work, to better constrain the processes controlling the effects of stratospheric ozone trends on tropospheric ozone.

**Code availability.** Information on the UM-UKCA configuration has been described in Archibald et al. (2020b). Due to intellectual property rights, we cannot provide the source code for the UM or the UKCA chemistry module. A number of research organizations and national meteorological services use the UM in collaboration with the UK Met Office to undertake basic atmospheric process research, produce forecasts, develop the UM code, and build and evaluate Earth system models. Further information on how to apply for a license can be found at https://www.metoffice.gov.uk/research/approach/modelling-systems/unified-model (last access: TS2). UM simulations are compiled in suites developed using the Rose suite engine (https://metomi.github.io/rose/doc/html/installation.html, Met Office & Contributors, 2020 TS3) and scheduled using the Cylc workflow engine (https://doi.org/10.5281/zenodo.7896205, Oliver et al., 2023 TS4). Both Rose and Cylc are available under version 3 of the GNU General Public License (GPL).

**Data availability.** Datasets used for this paper are publicly accessible and can be found at the links below: https://acd-ext.gsfc.nasa.gov/Data_services/cloud_slice/new_data.html (OMI-MLS, 2020 TS5); https://doi.org/10.5281/zenodo.1217184 (Hassler et al., 2018b TS6); https://doi.org/10.5067/LIS/LIS-OTD/DATA309 (Cecil and NASA MSFC, 2006 TS7); https://data.ceda.ac.uk/badc/toms/data/omi (last access: 17 May 2021, CEDA Archive, 2021).

**Supplement.** The supplement related to this article is available online at: https://doi.org/10.5194/acp-23-1-2023-supplement.

**Author contributions.** MRR performed the analysis of model and satellite data; BJK, BGL and RS provided support on the interpretation of satellite datasets and invaluable discussions on best practice in processing model data for comparison with satellite data; NLA performed model simulations; JW provided support on understanding the ozone sensitivity to isoprene emissions; JK and PTG implemented idealized tracers in the model; and JAP and ATA provided feedback and insight into the analysis.

**Competing interests.** The contact author has declared that none of the authors has any competing interests.

**Disclaimer.** Publisher's note: Copernicus Publications remains neutral with regard to jurisdictional claims in published maps and institutional affiliations.

**Acknowledgements.** Maria R. Russo, Nathan Luke Abraham, Barry G. Latter, Richard Siddans and Alex T. Archibald would like to thank NERC for financial support through the "North Atlantic Climate System Integrated Study" (NE/N018028/1). The RAL scheme to retrieve ozone height-resolved profiles (L2 data) from satellite UV sounders was developed through NERC's NCEO (award NE/R016518/1) and ESA's Climate Change Initiative. Production and analysis of gridded OMI (L3) data by RAL was

funded through NCEO. Alex T. Archibald, Nathan Luke Abraham and James Keeble thank the Met Office CSSP-China programme for funding the POzSUM project. Model integrations were performed on MONSooN2 (a collaborative high-performance computing facility funded by the Met Office and the Natural Environment Research Council) and NEXCS (a high-performance computing facility funded by the Natural Environment Research Council and delivered by the Met Office between 2017 and 2021). This work used JASMIN, the UK collaborative data analysis facility. Alex Archibald thanks the NERC PROMOTE project for funding (NE/P016383/1).

**Financial support.** This article was written with support from the UK National Environmental Research Council (NERC) through the long-term science programme on the North Atlantic Climate System Integrated study, ACSIS (grant NE/N018001/1).

**Review statement.** This paper was edited by Gabriele Stiller and reviewed by two anonymous referees.

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

## Remarks from the typesetter