# Peer review of "Seasonal, interannual and decal variability of Tropospheric Ozone in the North Atlantic: Comparison of UM-UKCA and remote sensing observations for 2005-2018"

_Atmospheric Chemistry and Physics, 2022_

## Author Comment (AC1)

Reply by the authors to Referee 1's comments on
"Seasonal, interannual and decal variability of Tropospheric Ozone in the North Atlantic:
Comparison of UM-UKCA and remote sensing observations for 2005–2018" (acp-2022-99)

Anonymous Referee 1 (RC1)

We wish to thank the Referees for their time and constructive comments, which have helped us improve the quality of the original manuscript. Our responses to these comments are provided below (the Referee's comments are shown in italic).

*The authors have presented a thorough analysis of their model simulation of tropospheric ozone's distribution and trends across the North Atlantic region, and have made very good use of satellite data. The paper is well written, the figures are clear and the topic is appropriate for the journal. However, I have three major concerns, described below, that should be addressed before the paper could be accepted for publication.*

*Major comments*

*1) Lines 124 – 135*

*This section suggests that the research community does not have a clear understanding of ozone trends across the North Atlantic region. However, there is a large body of evidence that ozone has increased over relatively long time periods, e.g. since the mid-1990s. In contrast, ozone's large interannual variability does make it difficult to detect trends on shorter time scales. I recommend that you provide a general overview of northern hemisphere ozone trends, highlighting what is known regarding long term trends, and then focus on the uncertainties of shorter trends across the North Atlantic region, as follows:*

*A) Begin with the latest assessment of global tropospheric ozone trends provided by IPCC AR6, based on observations (see Section 2.2.5.3 in Chapter 2 by Gulev et al., 2021) and models (Section 6.3.2.1 in Chapter 6 by Naik et al., 2021). Long-term trends since the mid-20th century are overwhelmingly positive (Tarasick et al., 2019) and trends since the mid-1990s are also positive in the free troposphere (Gaudel et al. 2020; Cooper et al., 2020; Chang et al., 2022), while trends at the surface and in the boundary layer are variable at northern mid-latitudes, but generally positive in the tropics (Cooper et al., 2020). This is nicely summarized by Figure 2.8 in Chapter 2. The most recent update to the OMI/MLS global trend can be found in the State of the Climate in 2020 report (Ziemke, 2021)*

*B) Recent papers by Cohen et al. (2018) and Gaudel et al. (2020) use IAGOS ozone profiles to show positive ozone trends in the upper troposphere of the North Atlantic and in the free troposphere above eastern North America and above Europe since 1994. At the surface ozone trends at remote sites show a range of positive and negative trends.*

*C) The new paper by Chang et al. (2022) combines all available IAGOS and ozonesonde profiles above western Europe to show ozone increased in the free troposphere from 1994 to 2019. Figure S12 in the supplement shows the trends are also positive on the shorter time scale of 2004-2019. But when the period is shortened to 2008-2019 the trends are much weaker. It's difficult to say if the weak 2008-2019 trend is simply due to a true weakening in*

*the ozone increase, of if the strong interannual variability across this short period introduces so much noise that the detection of a clear signal is not possible.*

We agree with the referee that ozone has increased over relatively long time periods and agree to implement changes as suggested in the referee's comments, with one small correction: IPCC, Chapter 6 is Szopa et al. not Naik et al. 2021. To address the referee's point we now start with a brief discussion on long-term ozone trends and have further expanded/clarified lines 124-135 as follows:

"Whilst there is consensus on the long term increase in global ozone burden, it is harder to pinpoint its magnitude due to the sparse nature and reliability of early ozone measurements. Using isotopic evidence from polar firn and ice and some model simulations, Yeung et al. (2019) estimated an ozone increase of less than 40% between 1850 and 2005. Tarasick et al. (2019) found surface ozone increases of 30-70% between historical (1877-1975) and present day (1975-2015) measurements at rural Northern Hemisphere stations; they also found that free tropospheric ozone has increased by ~50% between the same period for Northern Europe and the Eastern USA. CMIP6 model integrations are consistent with observations, with the multi-model ensemble mean producing an increase in tropospheric ozone burden of ~109 $\pm$ 25 Tg (~40%) between 1850–1859 and 2005–2014 (Szopa et al., 2021); this change in ozone has been attributed to an increase in anthropogenic ozone precursor emissions over the same time period (Szopa et al. 2021). In most recent decades, between the mid 1990s and present day, we see a more marked ozone increase in tropical regions compared to mid-latitudes (Gulev et al., 2021). At northern mid-latitudes, surface and low altitude ozone trends are variable, with some positive and some negative trends, but more positive values are observed in tropical regions (Cooper et al., 2020; Gaudel et al., 2020), where changes are between 2-17% per decade (Gulev et al., 2021). Similarly, ozone in the tropical free troposphere has increased more compared to ozone in the mid-latitude free troposphere, with increases of 2-12% per decade and 2-7% per decade, respectively (Cooper et al., 2020; Gaudel et al., 2020; Gulev et al., 2021; Chang et al., 2022).

Ozone trends in the North Atlantic can be influenced by a variety of factors. Anthropogenic emissions of ozone precursors have been decreasing in North America and Europe since the 1990s as a result of air quality policies; this reduction is potentially contributing to lower tropospheric ozone trends at northern mid-latitude compared to equatorial regions, where anthropogenic emissions of ozone precursors have continued to increase (Archibald et al., 2020a). Due to the relatively long lifetime of free-tropospheric ozone, 20-30 days (Young et al., 2013; Monks et al., 2015), North Atlantic ozone concentrations can also be affected by hemispheric transport of ozone generated by emissions outside of the local region (e.g., Butler et al., 2018; Sorooshian et al. (2020)). Other potential factors contributing to North Atlantic ozone trends include changes in tropical biogenic and biomass burning emissions, tropical $NO_x$ emissions from lightning and transport of ozone rich air from the stratosphere. Several studies have focused on ozone trends in Europe, USA and the North Atlantic region using surface measurements, sondes, aircraft and satellite observations (Cooper et al., 2014; Parrish et al., 2014; Oetjen et al., 2016; Heue et al., 2016; Gaudel et al., 2020; Cohen et al., 2018; Cooper et al., 2020; Chang et al., 2022). However, due to ozone's large interannual variability, calculated trends can be influenced by the reference years; furthermore, due to ozone spatial heterogeneity and large seasonal variations, reported trends can differ in

magnitude depending on the horizontal/vertical location and season (e.g. Cohen et al., 2018).

*2) I am puzzled by the apparent lack of any long-term ozone increase simulated by the model (which uses CMIP6 emissions), in contrast to the positive trends produced by the CMIP6 models (Skeie et al., 2020).  It would help if the authors can show a global map of modelled TCO ozone trends for the period 1992-2018 similar to the plot in the supplement for 2005-2018, so that the reader can understand where the model is and is not simulating ozone increases.  As shown by IPCC AR6 the strongest observed ozone trends have been in the tropics, especially for the period 1994-2016 when frequent IAGOS ozone profiles are available.  However, Figure 4 shows no ozone increase across the tropics for the period 1992-2018. Similarly, the model shows no ozone increase in either the tropical or the mid-latitude N. Atlantic regions for the period 2005-2018; overall the model indicates a weak ozone decrease. The authors could also compare their results to those of Zhang et al. (2016, 2021), who show that ozone increases in the tropics are driving ozone increases at northern mid-latitudes.*

First, we understand the referee's concern about the lack of trend in the model. It is true that our model integration between 2005 and 2018 underestimates the observed ozone trends in both tropical and North Atlantic regions. However, I would like to point out that the timeseries in Figure 4 (which extend back to 1992) are deseasonalised and 'detrended' anomalies (as pointed out in the caption and discussion) and therefore we would not expect to see trends in those plots. In the work by Skeie et al., it is clear that modelled ozone has increased between 1850 and 2010; however, UKESM shows the lowest increase amongst models which include tropospheric chemistry (Figure 3). It is also harder to pinpoint the sign and magnitude of the ozone trends from CMIP6 models in the more recent decades. Figure 2 in Skeie et al. shows that most models and observations have negative tropospheric ozone trends between 1980 and 2000, and a very small positive trend between 2000 and 2010, although there are large variations in magnitude between different models in both periods. This is further proof of the problems arising from calculating and comparing tropospheric ozone trends in the recent past, which partly arises from ozone's large interannual variability. To address the referee's point, we have produced a trend map for 1992-2018 (as suggested) which has now been included in supplementary Figure 7. As expected, this new panel shows generally positive trends in the Tropics and Asia, indicating that ozone between 1992 and 2018 has increased in the model and trends can vary greatly depending on the reference years considered.

*3) The reporting of trends in Table 2 needs to be revised in order to follow the advice of the American Statistical Association (ASA).*
*Table 2 reports zero when the calculated trend is less than the error of the trend estimate. This is the same as saying a trend is statistically insignificant when the p-value is greater than 0.05.  This method is no longer advised by the statistics community (ASA) and the authors should instead report all trend values and their uncertainty, as advised by the very influential paper by Wasserstein et al., 2019 (already cited 995 times, according to Web of Science). The readers can then make up their minds regarding the confidence they place on the trend value.  This method of reporting all trend values was adopted by TOAR (Chang et al., 2017; Tarasick et al. 2019) and is now being adopted by subsequent studies of ozone trends (Chang et al., 2020, 2021, 2022; Gaudel et al., 2020; Cooper et al., 2020; Thompson et al. 2021; see also Figure 2.8 in Chapter 2 of IPCC AR6 [Gulev et al., 2022])*

We thank the referee for this comment. The full data including trends and their uncertainty is already available in Supplementary Table 1, while we decided to report a percentage change over the period in question in Table 2; this was because we are comparing different chemical species and the absolute trend values would not give an indication on whether a trend is large or small. However, we have now aggregated these results so that Table 2 now provides absolute trend values and errors in addition to the percentage change of each species over the period in question. We have also removed supplementary Table 1 as no longer required.

*Minor Comments:*

*Line 48*

*Tying STT to 30 degrees latitude in the region of the descending branches of the Hadley and Ferrel cells is an over-simplification and not supported by the papers that are referenced. For example, Figure 12 of Yang et al. (2016) shows the latitude of peak STT varies with season, while, Figure 2 of Skerlak et al. (2014) shows the peak STT flux is typically in the 40-60 N latitude range.*

We agree with the referee that the statement was an oversimplification and have therefore changed the sentence to remove specific reference to 30 degrees latitude.

*Line 63*

*Tarrasick should be Tarasick*

This has now been corrected.

*line 109*

*When discussing transport of North American pollution into the North Atlantic Ocean, a good review is provided by Sorooshian et al. (2020).*

A reference to Sorooshian et al. (2020) has been added to the edited section (line 124-135) when discussing long range transport of ozone (see above).

*line 122*

*should jet speed by jet stream?  What is meant by ocean transports?*

"Jet speed" has been replaced with "speed of the jet stream". Ocean transport is a measure of the volumetric rate of transport by ocean currents. We have replaced lines 121-123 as follows:

"Significant decadal variability has been observed for the North Atlantic Oscillation and the speed of the jet stream (Hurrell, 1995; Woollings et al., 2015), ocean heat and salinity content (Robson et al., 2016; Reverdin, 2010), sea ice extent (Swart et al., 2015), and the rate of transport by ocean currents (Smeed et al., 2018)."

*Table 2*

*the trends are expressed as a percentage, but a percentage of what?*

We thank the referee for pointing out that the percentage had not been properly defined. This is the percentage change per decade, relative to the concentration of each species at the beginning of the period in question. This has now been clarified in the caption to Table 2.

---

## Author Comment (AC2)

Anonymous Referee 2 (RC2)

We wish to thank the Referees for their time and constructive comments, which have helped us improve the quality of the original manuscript. Our responses to these comments are provided below (the Referee's comments are shown in italic).

**General comment:**

*This paper evaluates changes in North Atlantic $O_3$ (2005-2018) using satellite observations and a chemistry-climate model, with a detailed analysis of the drivers of variability in the model and how this differs from observations. The abstract and introduction introduce the importance of the topic very clearly. The methods are well explained, the use of satellite data to derive $O_3$ column in vertical layers provides a very useful tool for model evaluation. There is thorough and detailed analysis throughout the study. The scope of the manuscript is certainly relevant to this journal.*

*Specific comments are listed below. I would recommend the manuscript for publication after these minor issues are addressed.*

**Specific comments:**

*Abstract: The abstract introduces the intent of the paper, methodology and major findings very well, but would benefit from including quantitative results. e.g. model/observation bias, trend in model $O_3$ vs observations, variability attributed to lightning NOx/STT.*

We have now included more quantitative information as suggested and modified the abstract as follows:

"Tropospheric ozone is an important component of the Earth System as it can affect both climate and air quality. In this work we use observed tropospheric column ozone derived from the Ozone Monitoring Instrument (OMI) and Microwave Limb Sounder (MLS) OMI-MLS, in addition to OMI ozone retrieved in discrete vertical layers, and compare it to tropospheric ozone from UM-UKCA simulations (which utilise the Unified Model, UM, coupled to UK Chemistry and Aerosol, UKCA). Our aim is to investigate recent changes (2005-2018) in tropospheric ozone in the North Atlantic region, and specifically its seasonal, interannual and decadal variability and to understand what factors are driving such changes. The model exhibits a large positive bias (greater than 5 DU or ~50%) in the Tropical upper troposphere: through sensitivity experiments, timeseries correlation and comparison with the LIS-OTD lightning flash dataset, the model positive bias in the Tropics is attributed to shortcomings in the convection and lightning parameterisations, which overestimate lightning flashes in the Tropics relative to mid-latitude. Use of OMI data, for which vertical averaging kernels and a priori information are available, suggests that the model negative bias (6-10 DU or ~20%) at mid latitudes, relative to OMI-MLS tropospheric column, could be the result of vertical sampling. Ozone in the North Atlantic peaks in spring and early

summer, with generally good agreement between the modelled and observed seasonal cycle. Recent trends in tropospheric ozone were investigated: whilst both observational datasets indicates positive trends of ~5 and ~10% in North Atlantic ozone, the modelled ozone trends are much closer to zero and have large uncertainties. North Atlantic ozone IAV in the model was found to be correlated to the IAV of ozone transported to the North Atlantic from the stratosphere (R = 0.77) and emission of NOx from lightning in the Tropics (R = 0.72). The discrepancy between modelled and observed trends for 2005-2018 could be linked to the model underestimating lower stratospheric ozone trends and associated stratosphere to troposphere transport. Modelled tropospheric ozone IAV is driven by IAV of tropical emissions of NOx from lightning and IAV of ozone transport from the stratosphere; however, the modelled and observed IAV differ. To understand the IAV discrepancy we investigated how modelled ozone and its drivers respond to large scale modes of variability. Using OMI height-resolved data and model idealised tracers, we were able to identify stratospheric transport of ozone into the troposphere as the main driver of the dynamical response of North Atlantic ozone to the Arctic Oscillation (AO) and the North Atlantic Oscillation (NAO). Finally, we found that the modelled ozone IAV is too strongly correlated to El Nino Southern Oscillation (ENSO) compared to observed ozone IAV. This is again linked to shortcomings in the lightning flashes parameterisation which underestimates/overestimates lightning flash production in the Tropics during positive/negative ENSO events."

*L103: Briefly expand on why the North Atlantic region is particularly important as well as the citation for more detail.*

We have now expanded lines 103-105 as follows:

"The North Atlantic is an interesting region where decadal changes in climate, spanning the atmosphere, ocean and cryosphere, interact to produce periods of faster warming and cooling, known as Atlantic multidecadal variability (AMV, Sutton et al., 2017). The AMV has been linked to a number of local and non-local impacts, ranging between rainfall anomalies, changes in the frequency of hurricanes and Greenland ice-sheet melt, to name just a few (Robson et al., 2018 and references therein). The leading mode of atmospheric variability in the North Atlantic climate system is the North Atlantic Oscillation (NAO), which drives interannual variability in tropospheric ozone, temperature and precipitation over Europe (Robson et al., 2018 and references therein). Understanding decadal changes in ozone and its drivers can help us predict future changes in North Atlantic ozone and how to mitigate its impact on, for example, exacerbating air quality problems."

*L131-134: Would benefit from clarifying exactly what the authors consider a "recent" trend. A number of studies, in particular TOAR assessments, have shown statistically significant increasing $O_3$ trends in the NH and in sites around the North Atlantic since the late 20th century (Gaudel et al., 2018, Tarasick et al. 2018). The authors rightly mention the uncertainties introduced by spatial and temporal inconsistency of these measurements but there is a broad consensus in the literature here.*

Similar concerns were raised by Reeree 1. We have therefore further expanded/clarified lines 124-135 as follows:

"Whilst there is consensus on the long term increase in global ozone burden, it is harder to pinpoint its magnitude due to the sparse nature and reliability of early ozone measurements. Using isotopic evidence from polar firn and ice and some model simulations, Yeung et al. (2019) estimated an ozone increase of less than 40% between 1850 and 2005. Tarasick et al. (2019) found surface ozone increases of 30-70% between historical (1877-1975) and present day (1975-2015) measurements at rural Northern Hemisphere stations; they also found that free tropospheric ozone has increased by ~50% between the same period for Northern Europe and the Eastern USA. CMIP6 model integrations are consistent with observations, with the multi-model ensemble mean producing an increase in tropospheric ozone burden of ~109 $\pm$ 25 Tg (~40%) between 1850–1859 and 2005–2014 (Szopa et al., 2021); this change in ozone has been attributed to an increase in anthropogenic ozone precursor emissions over the same time period (Szopa et al. 2021). In most recent decades, between the mid 1990s and present day, we see a more marked ozone increase in tropical regions compared to mid-latitudes (Gulev et al., 2021). At northern mid-latitudes, surface and low altitude ozone trends are variable, with some positive and some negative trends, but more positive values are observed in tropical regions (Cooper et al., 2020; Gaudel et al., 2020), where changes are between 2-17% per decade (Gulev et al., 2021). Similarly, ozone in the tropical free troposphere has increased more compared to ozone in the mid-latitude free troposphere, with increases of 2-12% per decade and 2-7% per decade, respectively (Cooper et al., 2020; Gaudel et al., 2020; Gulev et al., 2021; Chang et al., 2022).

Ozone trends in the North Atlantic can be influenced by a variety of factors. Anthropogenic emissions of ozone precursors have been decreasing in North America and Europe since the 1990s as a result of air quality policies; this reduction is potentially contributing to lower tropospheric ozone trends at northern mid-latitude compared to equatorial regions, where anthropogenic emissions of ozone precursors have continued to increase (Archibald et al., 2020a). Due to the relatively long lifetime of free-tropospheric ozone, 20-30 days (Young et al., 2013; Monks et al., 2015), North Atlantic ozone concentrations can also be affected by hemispheric transport of ozone generated by emissions outside of the local region (e.g., Butler et al., 2018; Sorooshian et al. (2020)). Other potential factors contributing to North Atlantic ozone trends include changes in tropical biogenic and biomass burning emissions, tropical NOx emissions from lightning and transport of ozone rich air from the stratosphere. Several studies have focused on ozone trends in Europe, USA and the North Atlantic region using surface measurements, sondes, aircraft and satellite observations (Cooper et al., 2014; Parrish et al., 2014; Oetjen et al., 2016; Heue et al., 2016; Gaudel et al., 2020; Cohen et al., 2018; Cooper et al., 2020; Chang et al., 2022). However, due to ozone's large interannual variability, calculated trends can be influenced by the reference years; furthermore, due to ozone spatial heterogeneity and large seasonal variations, reported trends can differ in magnitude depending on the horizontal/vertical location and season (e.g. Cohen et al., 2018).

*L302-304: O3 burden compares well to the observed values, but given the large overestimate in tropical TCO, this must be the result of negative bias elsewhere in the model, and therefore not indicative of good model performance relevant to the current study. Supplementary Figure 4f also supports this.*

We totally agree with the referee here, and in fact the next sentence expands on this point; lines 306-308 currently read:

"Archibald et al. (2020b) have shown that the UKCA global tropospheric ozone burden is consistent with observations as a result of an overestimate of TCO in the Tropics and an underestimate of TCO at mid latitudes, which is in line with our findings."

In order to make this clearer and avoid any possible misunderstandings, we have modified lines 300-308 as follows:

"Despite this spread in the observed TCO values, UKCA TCO, calculated for the same latitude band and period described in Gaudel (2018), shows values in the range 35-39DU, which are outside the range of uncertainty of the combined observations. Gaudel et al. (2018) reported a mean ozone burden, from 5 satellite datasets between $60^o$ S:$60^o$ N, of ~300 Tg +/- 6 % for the most recent satellite record (up to 2016). In our study the tropospheric ozone burden from OMI-MLS and UKCA for the 2005-2018 period are 297 and 301 Tg respectively. Although UKCA's ozone burden in the $60^o$ S:$60^o$ N range shows a very good agreement with observations, Archibald et al. (2020b) showed that the UKCA global tropospheric ozone burden is consistent with observations as a result of an overestimate of TCO in the Tropics and an underestimate of TCO at mid latitudes, which is in line with our findings (see also supplementary Figure 4)."

*Section 3.1: NOx emissions from soil and biomass burning also contribute to $O_3$ variability.*

We believe this Referee's comment is intended for Section 3.2 (not 3.1), as this is where the relationship between ozone and its precursor emissions is discussed. We agree with the Referee that NOx emissions from soil and biomass burning also contribute to O3 variability. In our model runs, these emissions are combined with anthropogenic NOx emissions and are referred in this section as 'surface NOx emissions'. We have now made this clearer and specifically acknowledged the importance of these extra sources by modifying lines 356-361 as follows:

"Present day anthropogenic emissions are generally well constrained, and their geographical locations, seasonal variations and magnitudes are derived from emission inventories and inverse modelling techniques (Lamarque et al., 2010; Feng et al., 2020). In contrast, some natural emissions of ozone precursors can have quite large uncertainties; these include CO and NOx from biomass burning, soil NOx, biogenic isoprene and NOx from lightning. An overestimate of such ozone precursors emissions in the model could therefore result in an overestimate of tropospheric ozone. Please note that, with the exception of lightning, all other natural and anthropogenic sources of NOx are combined in the model and referred to as surface NOx emissions."

*Section 4.1: More context from a modelling perspective would be very informative here. How does the UM-UKCA compare to other relatable modelling studies? Is the lack of an O3 trend a consistent problem across CCMs (if so why?) or is it just UM-UKCA?*

To address the Referee's comment we have now added the following paragraph to lines 505-506 in section 4.1:

"In contrast to observed trends, UKCA ozone trends tend to be much smaller in magnitude and effectively zero (within the error) for both the tropical and mid latitude part of the domain. Skeie et al. (2020) investigated ozone trends in CMIP6 model simulations; while it is clear that modelled ozone has increased significantly between 1850 and 2010, it is hard to pinpoint the sign and magnitude of the ozone trends from CMIP6 models in the more recent decades. Figure 2 in Skeie et al. shows that observed ozone trends for 2000-2010 are less than 5% per decade, while modelled trends for the same period show many models have trends very close to zero, and generally within ± 2% per decade. Although the period we are investigating (2005-2018) is not the same as shown in Skeie et al. (2020), their findings suggest that UKCA ozone trends in the more recent decades are comparable to other CMIP6 models and that accurately estimating tropospheric ozone trends over relatively short time periods remains a challenge due to ozone's large interannual variability."

***Technical corrections:***

*L122: Jet stream?*

"Jet speed" has been replaced with "speed of the jet stream".

*Figure 3: If the shaded area is of no interest in all 4 panels perhaps remove it from the figure?*

We would prefer to keep the figure as it is for consistency, as all other figures show the full North Atlantic domain. Also, although not specifically within the upper troposphere, comparison of model and satellite over the shaded area provides some further information for the reader regarding model biases higher up in the atmosphere.

*Figure 8. No label on y-axis.*

We thank the Referee for pointing this out. The label on the y-axis should be "Ozone anomalies (DU)". This has now been corrected.

*Table 2. Unit. % change per year or over whole period?*

We thank the referee for pointing out that the percentage had not properly defined. This is the percentage change per decade, relative to the concentration of each species at the beginning of the period in question. This has now been clarified in the caption to Table 2.

*Figure 9. The black boxes next to the shaded area don't clearly highlight the area of interest. Changing the colour of boxes/shading could improve this so it's easier to pick out the important regions.*

The colour of the black boxes has now being changed to magenta and lines were made thicker to address the Referee's comment.

*Supplementary Figure 8. No label on y-axis.*

We thank the Referee for pointing this out. The label on the y-axis should be "Ozone anomalies (DU)". This has now been corrected.